# Building block aspect ratio controls assembly, architecture, and mechanics of synthetic and natural protein networks

Matt D. G. Hughes [1], Sophie Cussons [2,3], Benjamin S. Hanson [1],
Kalila R. Cook [1], Tímea Feller[4], Najet Mahmoudi [5], Daniel L. Baker [1],
Robert Ariëns [4], David A. Head [6], David J. Brockwell [2,3] &
Lorna Dougan [1,2] ✉

Fibrous networks constructed from high aspect ratio protein building blocks are ubiquitous in nature. Despite this ubiquity, the functional advantage of such building blocks over globular proteins is not understood. To answer this question, we engineered hydrogel network building blocks with varying numbers of protein L domains to control the aspect ratio. The mechanical and structural properties of photochemically crosslinked protein L networks were then characterised using shear rheology and small angle neutron scattering. We show that aspect ratio is a crucial property that defines network architecture and mechanics, by shifting the formation from translationally diffusion dominated to rotationally diffusion dominated. Additionally, we demonstrate that a similar transition is observed in the model living system: fibrin blood clot networks. The functional advantages of this transition are increased mechanical strength and the rapid assembly of homogenous networks above a critical protein concentration, crucial for in vivo biological processes such as blood clotting. In addition, manipulating aspect ratio also provides a parameter in the design of future bio-mimetic and bio-inspired materials.

Biology exploits geometry over a wide range of length scales, from honeybees hexagonally packing their hives to ensure the maximum honey storage for the minimum expenditure of wax[1] on the bulk scale, to the gyroid structures in the blue wings of the *Morpho didius* butterfly on the nano-scale, which only scatter light of a particular wavelength regardless of incident angle[2]. As living systems often exploit geometry to achieve functionality with minimal energetic cost[3–5], investigating whether hierarchical protein assemblies are also geometrically optimised could provide routes for the development of future bio-mimetic and bio-inspired materials.

Aspect ratio (AR) is a key property in geometry, often defined as the ratio between the length and width of an object. Fibrous assemblies of high AR biomolecules, such as the staggered twisted structures of collagen[6,7] and fibrin fibrils[8,9], are ubiquitous in all living systems[10–13]. Their ubiquity suggests biomolecule AR may be an evolutionarily optimised solution[14,15]. Knowledge of the impact of building block AR on hierarchical biomechanics is important to understand the fundamental design principles of nature's networks and could reveal routes for the design of biomaterials. High AR molecules have a length which is many times longer than the width of the molecule, with examples including carbon nanotubes (CNTs)[16,17], cellulose nanocrystals[18,19] and

[1]School of Physics and Astronomy, Faculty of Engineering and Physical Sciences, University of Leeds, Leeds, UK. [2]Astbury Centre for Structural Molecular Biology, University of Leeds, Leeds, UK. [3]School of Molecular and Cellular Biology, Faculty of Biological Sciences, University of Leeds, Leeds, UK. [4]Leeds Institute of Cardiovascular and Metabolic Medicine, Faculty of Medicine and Health, University of Leeds, Leeds, UK. [5]ISIS Neutron and Muon Spallation Source, STFC Rutherford Appleton Laboratory, Oxfordshire, UK. [6]School of Computing, Faculty of Engineering and Physical Science, University of Leeds, Leeds, UK. ✉e-mail: L.Dougan@leeds.ac.uk

elastomers[20,21]. Recent work on carbon nanotubes and elastomers has attempted to understand the role of AR in the percolation and adhesion of synthetic fibre networks. A comparative study of CNTs found that higher AR CNTs percolated into more rigid assemblies than low AR CNTs[22]. However, a similar study using the bulk macro-scale AR of a complete elastomer fibre found that lower AR elastomer fibres resulted in threefold stiffer networks than higher AR fibres[23]. Despite the research in synthetic systems and the ubiquity of these high AR networks in nature, comparatively little is known about the role of nanoscale biomolecule AR on the mesoscale formation, mechanics and structure of their networks. This, in part, is due to the complexity of biological protein networks which often contain multiple components of differing geometric, chemical and physical properties. In order to understand the distinct role of building block AR in protein networks, we need to take a bottom-up approach by engineering a controllable model system.

Here, we exploit protein engineering to provide direct control of the AR of the nanoscale building block. A suite of engineered folded homo-polyprotein building blocks (comprised of one to seven copies of the same single domain protein joined by short 4-residue linkers) with a range of ARs was created as a model experimental system to determine the impact of AR on protein network properties. Hydrogels constructed from each building block were analysed by a multiscale experimental approach, including rheology and small-angle neutron scattering (SANS), to observe the changes in the mechanics and network topology as AR was varied. In conjunction, computational simulations were performed to support and supplement our experimental results. We used a fibrin network as a model naturally occurring network and performed a similar rheological analysis to extract the importance of AR in natural networks. This multi-protein and multimodal cross-length scale approach enables us to demonstrate that AR has a significant effect on the formation, topology and mechanics of protein networks.

## Results

### Selection of model protein system

Polyproteins, tandem repeats of single protein domains (Fig. 1a), are ideal hydrogel building blocks[24,25] as the number of domains in a polyprotein and the domain linker regions can be easily controlled[26–28], providing opportunities to investigate the effect of building block AR on protein networks. Protein L (pL) was selected as the model domain due to its well-characterised single molecule properties[29–31]. Our model protein L is a 64-residue antibody binding domain (Fig. 1a) from *Finegoldia magnu* with an I11Y mutation introduced into pseudo-wild-type Y47W protein L[29] (Methods). The variant used in this study has four surface exposed tyrosine residues as critical coordination of four is required to form a continuous self-supported network in an athermal frictional system[32,33] (Note: 2D thermal simulations have been performed[34], which demonstrated that a significant increase in rigidity of the simulated network was observed at critical coordination despite the inclusion of thermal fluctuations). Tyrosine residues are essential for the residue-specific photo-chemical crosslinking method that we employ in this work to form our pL hydrogel networks. To confirm our pL polyprotein constructs were in a folded state, we performed circular dichroism (CD) spectroscopy measurements on our polyprotein constructs. Supplementary Fig. 1 shows that the spectra for all the pL polyproteins have the same profile as the pL monomer, demonstrating that the pL polyproteins are in a folded conformation, thus allowing us to control the aspect ratio by controlling the number of tandem repeats.

Polyproteins with short linker (~5aa) regions between neighbouring folded domains are very stiff, so they behave as rod-like objects[35,36]. For this reason, our pL polyprotein constructs were designed with extremely short linker regions (four residues between neighbouring domains (supplementary information)) to ensure that differences in network properties were due to geometric differences between constructs as opposed to flexibility. We perform small-angle

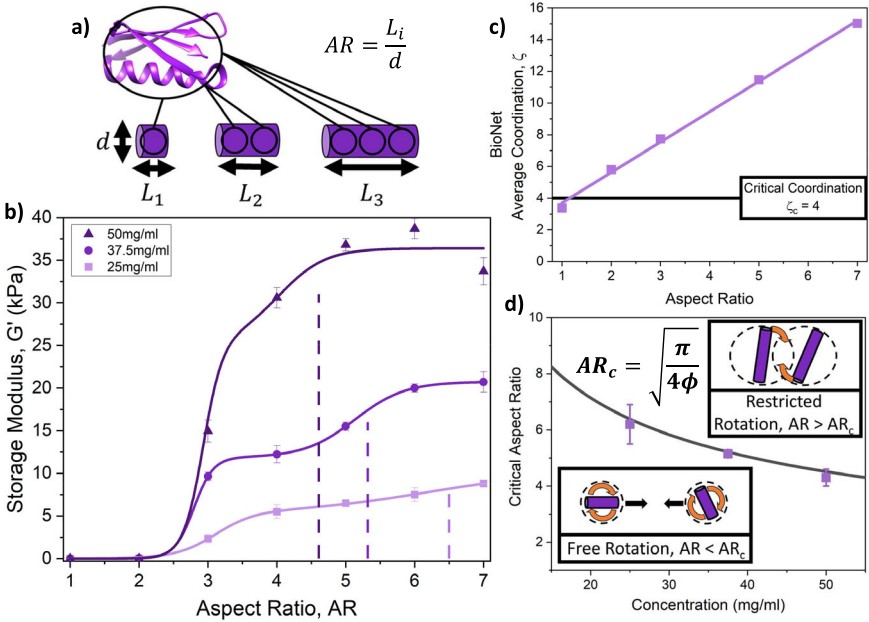

**Fig. 1 | Aspect ratio controls the coordination and rotational restriction of the building block, altering the protein network rigidity. a** Crystal structure of protein L (PDB code: 1HZ6) and a schematic depicting the increasing aspect ratio (AR) of polyprotein building blocks as an increase in the polyprotein length. **b** The storage modulus of pL hydrogels as a function of building block AR, at varying pL polyprotein concentrations (measured in mg ml⁻¹), solid lines show fits using a double sigmoidal fit (Eq. 4), full fitting parameters provided in Supplementary Table 2. Dashed lines show the extracted $AR_c$ points for each protein concentration.

Supplementary Table 3 shows the equivalent protein and water volume fractions of pL hydrogels. Data points are presented as mean values ± SEM, where $n = 3$. **c** Average coordination, ζ, determined from the simulation suite BioNet, as a function of building block AR. **d** $AR_c$ values extracted from panel **b** using Eq. 4 (purple squares) and the calculated (Eq. 1) critical aspect ratio (black line) of pL polyprotein rods in solution (i.e. AR at which the rod-like polyproteins can no longer freely rotate in solution) as a function of polyprotein concentration in solution. All error bars show the standard error.

X-ray scattering to confirm our rationale regarding the rigidity of our polyproteins. The scattering curve of pL$_7$ (our longest polyprotein, i.e. the most likely to be flexible) shown in Supplementary Fig. 2, exhibits a Porod exponent of approximately 1, which is indicative of a rod or a polymer in its fully extended confirmation[37], which suggests that our polyproteins are rigid and can be modelled as rod-like.

## Modulation of network rigidity

To understand the impact of building block AR on the mechanics of protein networks, we employ pseudo-strain-controlled oscillatory rheology to characterise the linear mechanical behaviour of our photo-chemically cross-linked (methods) pL hydrogel networks. Figure 1b shows the storage modulus of the pL hydrogels at 1 Hz extracted from frequency sweeps (Supplementary Fig. 3) as a function of the building block AR and protein concentration. At low building block AR, (AR = 1,2) G′ is ~0 kPa; once AR is increased to approximately three, the G′ increases for all concentrations and continues to increase up to a plateau as AR increases further. Additionally, as the concentration of protein increases, we observe an enhanced rigidity of the networks that are formed, likely due to the additional load-bearing material present in the system. To allow for the comparison of proteins with different building block aspect ratios and their impact on protein network formation, the protein volume fraction and water volume fraction are known fixed values and are unchanged throughout the experiments. The protein volume fraction/concentration in mg ml$^{-1}$ (and therefore water volume fraction) is the same at the start and end of the experiment, with no water uptake or evaporation (see Methods). The protein and water volume fractions are listed in Supplementary Tables 3 and 4. The low G′ values at low ARs suggest that the 1-mer and 2-mer do not form self-supporting networks. This is confirmed by the loss ratio (defined as a ratio of the loss modulus to the storage modulus) (Supplementary Fig. 4), which is above one for low AR samples (AR < 3) exhibiting fluid-dominated behaviour as opposed to the viscoelastic solid-like behaviour expected for a gel. For gels constructed from building blocks of AR three and greater, we observe loss ratios of approx. 0.03 demonstrating that these gels are self-supporting networks which are dominated by their elastic behaviour. We perform thermogravimetric analysis to confirm there is no significant change in the water content as a function of building block AR. Supplementary Fig. 5 shows that the weight loss is invariant with respect to AR, meaning there is no significant change in water content as a function of AR, i.e. changes in water content is not sufficient to explain the observed changes in rigidity. A possible reason for this transition to a self-supporting network as AR is increased is that the network is under-coordinated at low ARs, i.e. the average coordination, ζ, of each polyprotein building block, is lower than 4, so it would likely not be able to form a self-supporting gel network.

Coarse-grained simulations were performed using the simulation platform BioNet[36,38,39] (Methods) to investigate the coordination of network (polyprotein) building blocks of different ARs. Figure 1c shows how the average coordination of each polyprotein building block, ζ, varies as a function of AR, displaying a linear relationship. From the graph, the average coordination of the simulated pL systems is below critical coordination at an aspect ratio of 1 but quickly passes critical coordination as AR is increased. These results suggest that while AR is a method of controlling the coordination of individual building blocks, it is not sufficient to explain the sudden increase in G′ observed when AR = 3. Previous literature[40,41] on fibrous protein networks has observed that the rigidity and mechanical behaviour of heterogeneous protein networks are governed by a combination of building block coordination and connectivity (i.e. branching) between bundled protein fibres. Similarly, it has been observed by Del Gado et al.[42,43] that in heterogeneous clustered colloidal networks, only the connections between clusters are significant in defining rigidity (i.e. the highly coordinated particles inside clusters do not contribute

significantly to the rigidity). Folded protein hydrogels have been observed to exhibit highly heterogeneous clustered structures[44–46], so we would expect that the formation of a self-supporting network would be governed by a combination of building block coordination and connectivity between the clusters of proteins.

The data in Fig. 1b is fitted with a dual-sigmoid function (Methods, Eq. 4). We show that a single sigmoid (Supplementary Fig. 6), modelling the system as undergoing a single transition beyond the ζ$_c$ as AR is increased, is not sufficient to fit the G′ vs. AR trend. Hence the successful dual-sigmoid fit suggests that two mechanisms are responsible for the increase in G′ with AR. The first of these mechanisms has already been discussed above and is likely due to a combination of building block coordination and system branching/cluster connectivity which enables the formation of a self-supporting network. The second mechanism may be due to the restriction of free rotation of the polyprotein building block during network formation, i.e. a polyprotein construct in solution is unable to rotate about its centre of mass without colliding with another polyprotein in the solution. To demonstrate this, we derive the free rotation limit of a rigid cylindrical rod[38] (Eq. 1, Supplementary Information), also known as the transition from the dilute to the semi-dilute regime for rod-like particles[47]

$$AR_{crit}^{rod} = \sqrt{\frac{\pi}{4\phi}} \tag{1}$$

Where $AR_{crit}^{rod}$ is the critical AR for a mono-dispersed solution of rods at a volume fraction of $\phi$, above which the rods can no longer freely rotate about their centre of mass without colliding with one another. Fitting a dual sigmoid to the data in Fig. 1b allows a value for the critical aspect ratio, $AR_c$, to be extracted. We would expect $AR_c$ to correlate with $AR_{crit}^{rod}$ (Eq. 1) if the second mechanism is due to restricted rotational motion. Figure 1d shows the extracted $AR_c$ values for each protein concentration, superimposed with the theoretical curve of $AR_{crit}^{rod}$ against protein concentration (Eq. 1), demonstrating that by increasing the protein concentration, the critical aspect ratio for rotational limit is decreased. This result shows the extracted $AR_c$ values correlate extremely well with Eq. 1, demonstrating that the restriction of free rotation is the second mechanism that defines the G′ dependency on the AR of the network building block.

## Transition in network topology

We have shown that as the building block increases in AR, the assembly of the hydrogel network shifts from being dominated by translational diffusion (i.e. building blocks must translate in space in order to collide and form cross-links) to being dominated by rotational diffusion (i.e. building blocks can collide and cross-link only via rotation about their centre of mass). To investigate the structural changes due to the restriction of free rotation of high AR building blocks when AR > $AR_c$, we use SANS to probe the network structure of pL hydrogels (Fig. 2).

Figure 2b and Supplementary Fig. 7a show the scattering curves of pL hydrogels with varying building block ARs at 50 mg ml$^{-1}$ and 25 mg ml$^{-1}$ protein concentrations, respectively. These curves show three key features; (i) at high q, a small 'shoulder' in the I(q) signal is observed corresponding to the size of the individual domains in the polyprotein constructs (width, d), which is the same for all the building blocks; (ii) in the mid-q region the pL hydrogels exhibit a decreasing power law with increasing q; (iii) finally at low q a turn over to plateau is seen (which is more prominent in 50 mg ml$^{-1}$ samples than 25 mg ml$^{-1}$ samples, compare Fig. 2b and Supplementary Fig. 7a). Close inspection of the I(q) curves shows the maximum intensity at low q values decreases with increasing AR, suggesting differences in the structure of each protein hydrogel. Previous characterisation of folded protein hydrogels has identified fractal-like structures present in the network architecture[45,46,48,49]. Fitting the curves in Fig. 2b and S6a with Eq. 5 and modelling the scattering of both the individual protein building blocks

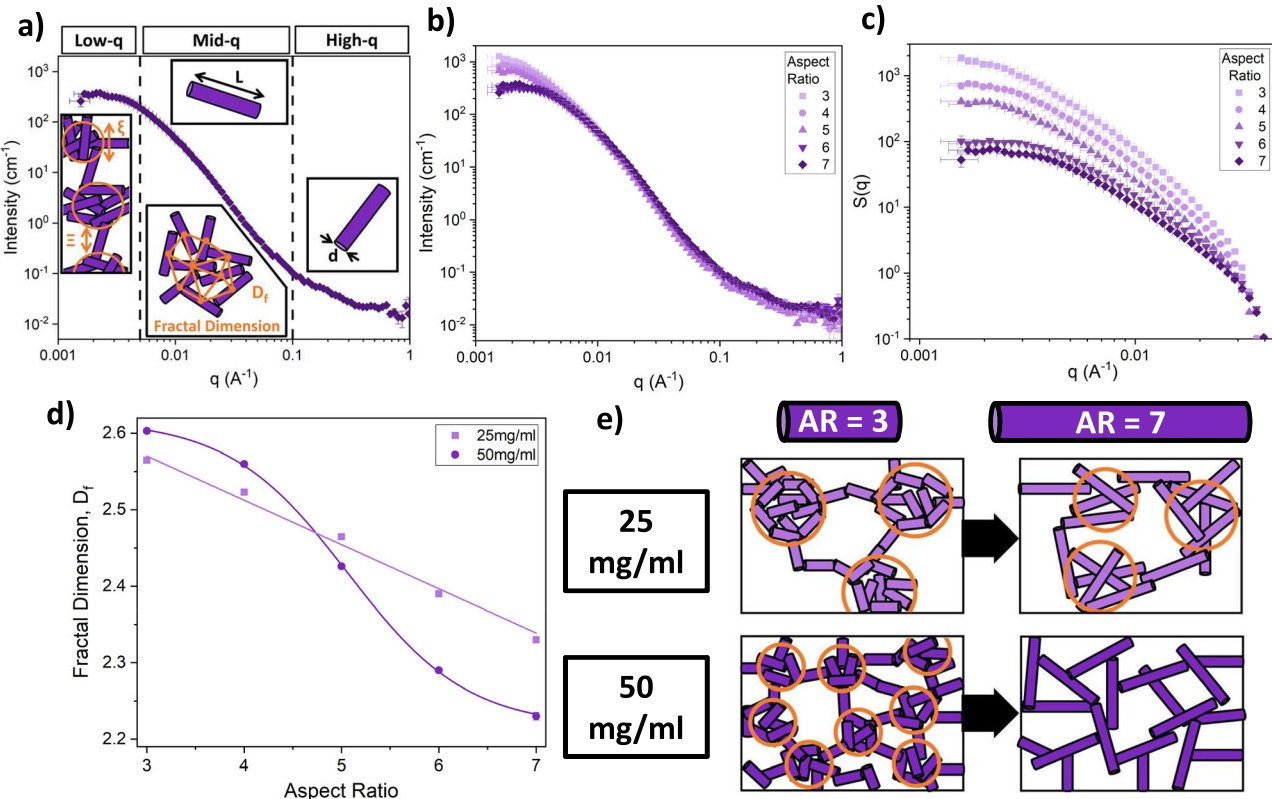

**Fig. 2 | Small-angle scattering reveals two distinct network architectures accessed by altering the building block aspect ratio. a** Schematic depicting the different q-regions of SANS curves and the potential information that can be extracted. Including low $q$ (characteristic network length scale, $\xi$ or $\Xi$); mid-$q$ (building block length, $L$, and network geometry, i.e. fractal dimension, $D_f$); and high-q (building block width, $d$). **b** SANS curves for pL hydrogels as a function of building block aspect ratio (AR) at a protein concentration of 50 mg ml$^{-1}$. Data points are the mean values of logarithmically binned histograms at each $q$-value; the $y$ error bars show the standard error, while the $x$ error bars are related to the detector resolution. The number of independent sample repeats, $n = 1$. **c** Extracted

structure factor for pL hydrogels as a function of building block AR at protein concentrations of 50 mg ml$^{-1}$. The error bars show the propagated errors from Fig. 2b. **d** Extracted fractal dimension of pL hydrogels as a function of building block AR at protein concentrations of 25 mg ml$^{-1}$ (light purple) and 50 mg ml$^{-1}$ (dark purple). The error bars here denote the fitting error of the fractal dimension to SANS curves. Solid lines show a linear (light purple) and sigmoid (dark purple) fit to the 25 and 50 mg ml$^{-1}$ data, respectively. Supplementary Table 3 shows the equivalent protein and water volume fractions of pL hydrogel. **e** Schematic representation of structural changes as building block AR is increased while at a constant concentration of 25 mg ml$^{-1}$ (light purple) and 50 mg ml$^{-1}$ (dark purple).

and the mesoscale network architecture allows us to extract information about the structure of the hydrogel network. To clearly illustrate the difference in structure between hydrogels constructed from building blocks with different ARs at 50 and 25 mg ml$^{-1}$, we plot the extracted structure factor, $S(q)$, (Eq. 6) in Fig. 2c and Supplementary Fig. 7b, respectively. A power law trend of $S(q)$ with increasing q is observed in the mid-$q$ region, indicative of the fractal dimension of the network architecture. Furthermore, the $S(q)$ curves show a decrease in this power law's exponent with increasing AR, suggesting a change in the network architecture. The structure factor (Eq. 6) contains key structural parameters such as the fractal dimension of the network, $D_f$, and the characteristic length of the network structure, $\xi$. In addition to $S(q)$, we can extract the proportion of protein in the fractal network, $P_c$, from Eq. 5.

Figure 2d and Supplementary Figs. S8 and S9 show how $D_f$, $\xi$ and $P_c$, respectively, vary as a function of polyprotein AR at protein concentrations of 25 and 50 mg ml$^{-1}$. As AR is increased, the $D_f$ of the network is decreased at both protein concentrations. This result is to be expected as the $D_f$ is a measure of the space-filling capacity of an object, and it is intuitive that sticky higher AR building blocks that are diffusing and linked together randomly will form much more porous structures (due to the elongated dimension). Such structures will be less space-filling (Fig. 2e) and hence have lower $D_f$ compared to structures formed by low AR building blocks. Interestingly, while at a concentration of 25 mg ml$^{-1}$ the protein network appears to show a

linear decrease in $D_f$ with AR is shown, while at 50 mg ml$^{-1}$, a clear sigmoid shape is observed, suggesting a transition between two structural regimes as AR is increased. To confirm these trends observed in the fractal dimension are not a result of the model chosen, we also perform model-independent Guiner–Porod fits to the low/mid-$q$ region in order to extract the Porod exponent (i.e. the power law exponent). Supplementary Fig. 10 shows similar trends in the Porod exponent as a function of AR that are exhibited by the fractal dimension values in Fig. 2d. From these results, we suggest that the structure of pL hydrogels at 50 mg ml$^{-1}$ transitions from a collection of fractal-like clusters of cross-linked proteins to a homogeneous network of cross-linked polyprotein rods (Fig. 2e). This interpretation is supported by the extracted $P_c$ values (Supplementary Fig. 9) which show that the proportion of protein in the fractal network increases in a sigmoidal fashion from $0.487 \pm 0.002$ at AR = 3 to $0.745 \pm 0.003$ at AR = 7. This dramatic increase in the proportion of protein in the fractal network is indicative of having little to no protein connecting "clusters" together and instead suggests all proteins are part of a homogeneous "mono-cluster" which spans the whole system. Interestingly, a similar trend is not observed in $\xi$ as a function of AR (Supplementary Fig. 8), instead samples at 50 mg ml$^{-1}$ show no significant change in $\xi$ (~280 Å) as AR is increased, while those at 25 mg ml$^{-1}$ exhibit a linear decrease from ~700 Å at AR = 3 to ~500 Å at AR = 5.

Fitting the $D_f$ results at 50 mg ml$^{-1}$ with a sigmoid function allows us to extract the midpoint of the transition as AR$_{midpoint} = 4.9 \pm 0.3$,

which is in reasonable agreement of $AR_{crit}^{rod}$ (predicted with Eq. 1) of 4.52. Additionally, we also fitted a sigmoid to the $D_f$ results at 25 mg·ml⁻¹ (Supplementary Fig. 11) and find an $AR_{midpoint}$ of 6.5 ± 0.3, which also agrees with $AR_{crit}^{rod}$ at 25 mg ml⁻¹ (predicted with Eq. 1) of 6.37. These results suggest that the transition of network formation from dominantly translational diffusion to dominantly rotational diffusion is altering and defining the structure of the pL hydrogel networks.

To investigate if rotation is a key component to the alteration of the hydrogel topology, we use a kinetic lattice model (Methods) to simulate the formation of colloidal networks of building blocks with different ARs. Importantly, the kinetic lattice model does not include rotational diffusion, so we would not expect to observe the sigmoidal trend in $D_f$ at 50 mg ml⁻¹ if rotational diffusion was the driving mechanism. The extracted $D_f$ values from the simulations (Supplementary Fig. 12) show a linear trend with increasing AR at both 25 mg ml⁻¹ and 50 mg ml⁻¹. These results confirm that translational diffusion alone is not sufficient to produce the transition in network structure and thereby suggesting that rotational diffusion is a key mechanism driving the architecture of protein networks. Interestingly, the gradient of $D_f$ vs. AR for the 25 mg ml⁻¹ simulations (−0.045 ± 0.002) is remarkably close to the gradient extracted from the SANS values in Fig. 2d (−0.058 ± 0.003), suggesting that when $AR < AR_c$ translational diffusion dominates formation of the protein network.

## Free rotation limit in hydrogel network formation

To investigate the importance of translationally diffusion limited (TDL) and rotationally diffusion limited (RDL) formation in determining network architecture we investigate how the lag time of hydrogels constructed from pL₇ (Fig. 3a) varies as function of pL₇ volume fraction (or concentration). The lag time, $\tau$, is defined here as the time from initiation of cross-linking until a mechanical response is measured (Supplementary Fig. 13a), demonstrating a percolated rigid network has formed. During gelation we note that there is an increase in the normal force in the negative direction (Supplementary Fig. 14) suggesting that the gels are undergoing slight deswelling. The extracted $\tau$ values for pL₇ for varying volume fractions, $\phi$, of pL₇ are plotted in Fig. 3b.

At low $\phi$ values (<3%), $\tau$ decreases with increasing $\phi$, as a higher concentration of the building block improves the probability that two will translationally diffuse and collide to form a crosslink. However, this trend breaks down as $\phi$ further increases and a levelling out of $\tau$ at higher $\phi$ is observed ($\tau \sim 4$ s at $\phi > 3\%$). If the $\tau$ does not continue to decrease as $\phi$ increases, then this suggests that translational diffusion is not the sole mechanism governing the network formation of pL₇ hydrogels. We propose that this additional mechanism is the rotational diffusion of the pL₇ building blocks, and to determine if this is the case, we fit the data in Fig. 3b with Eq. 2.

$$\tau = \frac{\tau_t}{\phi} + \tau_r \qquad (2)$$

Where $\tau_t$ and $\tau_r$ are characteristic time scales relating to translational and rotational diffusion, respectively. In Eq. 2, there are 2 terms; the first term models translational diffusion and is inversely proportional to $\phi$ (the mean free path of 2 diffusing particles is inversely proportional to $\phi$[50]), and the second term models rotational diffusion and is

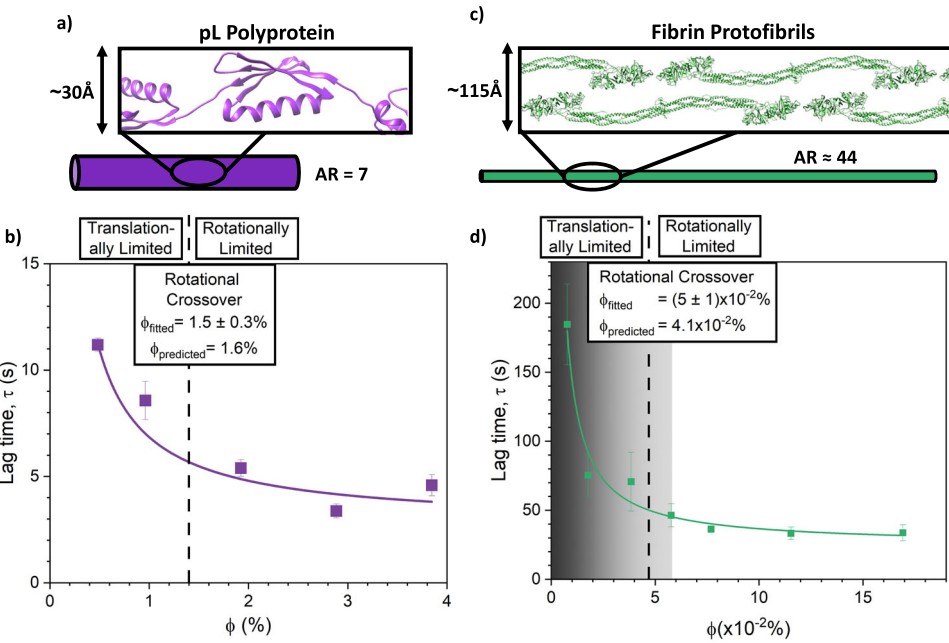

**Fig. 3 | Increasing aspect ratio (AR) shifts the dominant network formation mechanism of synthetic (pL hydrogels) and natural (fibrin) protein networks from translational to rotational diffusion. a** Schematic representation of the pL₇ building block, which has an aspect ratio of 7, with an inset of the pL₇ structure as predicted by AlphaFold[77]. **b** Network formation lag time, $\tau$, extracted from gelation curves (example shown in Supplementary Fig. 13a) as a function of protein volume fraction for pL₇ hydrogel networks. Supplementary Table 4 shows the protein and water volume fractions of each pL₇ lag time sample. Data points are presented as mean values ± SEM, where $n = 3$. **c** Schematic representation of fibrin protofibrils, which have an approximate aspect ratio of 44 (20−25 monomer units)[78], with inset of the crystal structure of human fibrinogen (PDB code: 3GHG) arranged in an offset stacked structure. **d** Network formation lag time, $\tau$, extracted from gelation curves

(example shown in Supplementary Fig. 13b) as a function of protein volume fraction for fibrin networks. Supplementary Table 4 shows the protein and water volume fractions of each of the fibrinogen lag time samples. Solid lines (in panels (**b**) and (**d**)) show the fits using Eq. 2, while dotted lines show the extracted values for the rotational crossover volume fraction, i.e. the volume fraction when the building blocks are no longer able to freely rotate, calculated using Eq. 3 (equivalent fibrin concentration of 0.7 ± 0.1 mg ml⁻¹). Additionally, the predicted values from Eq. 1 are shown for comparison. The grey region in panel **d**) shows the range of fibrinogen volume fractions below 0.75 mg ml⁻¹ corresponding to critically low blood fibrinogen concentrations, i.e. hypofibrinogenemia. Data points are presented as mean values ± SEM, where $n = 3$.

independent of $\phi$ (the rotation rate of rod-like particles is not dependant on $\phi$ in the dilute regime[47,51]). By equating the two terms on the right of Eq. 2, the volume fraction at which both mechanisms equally contribute to the $\tau$ behaviour, $\phi_{crit}$, can be determined, where

$$\phi_{crit} = \frac{\tau_t}{\tau_r} \tag{3}$$

Fitting the curve in Fig. 3b with Eq. 2 and calculating $\phi_{crit}$ to be $(1.5 \pm 0.3)\%$, we see that this value is remarkably similar to the $\phi_{crit}$ value that is predicted by Eq. 1, i.e. 1.6%. This demonstrates that there is a transition in the dominant formation mechanism of pL hydrogel networks. Furthermore, it suggests that the transition is from a TDL formation to an RDL formation as AR increases, such that $AR > AR_c$ is expected from the SANS results. To confirm that there is a transition from the dilute (i.e. TDL) to the semi-dilute (i.e. RDL) regime in $pL_7$ solutions, we perform dynamic light scattering (DLS) experiments on $pL_7$ solutions at the same volume fraction as in Fig. 3b. DLS probes the dynamics of our protein building blocks in solution, allowing us to extract the translational diffusion constant, $D$, as a function of volume fraction (Supplementary Fig. 15)[52,53]. As the volume fraction of $pL_7$ is increased, we observe a sigmoidal transition in $D$ from $\sim 4 \times 10^{-11}$ m² s⁻¹ to $\sim 3 \times 10^{-11}$ m s⁻¹, demonstrating that above this volume fraction, there is a shift in the dynamical behaviour of the protein in solution. This transition is centred at a volume fraction of $2 \pm 0.2\%$ (Supplementary Fig. 15b), which is within error of the fitted value extracted from our lag time data (Fig. 3b) and is in close agreement with the value predicted by our geometric model of 1.6% (Eq. 1). From Doi-Edwards theory[47] we would expect a reduction in $D$, if the system were shifting from the dilute to the semi-dilute regime, due to a restriction in translational diffusion perpendicular to the polyprotein rod-axis[54].

These results demonstrate that there is a transition in the dynamics of $pL_7$ solutions which is responsible for a shift in the dominant network formation mechanism resulting in more rapid gelation (Fig. 3b), more homogeneous networks (Fig. 2d, e) and more rigid networks (Fig. 1b). Furthermore, this transition coincides with the value predicted by our geometric rotation model (Eq. 1) suggesting that that shift in formation mechanism in these protein networks is from TDL to RDL.

### The emergence of free rotation limit in living systems

The AR of a pL building block has a critical role in defining the dominant assembly mechanism, structure and subsequent rigidity of pL hydrogel networks due to a crossover from TDL formation to RDL formation. We next investigated if such a crossover occurs in naturally occurring fibrous networks using fibrinogen as a model protein (Fig. 3c). Fibrinogen is an abundant large glycoprotein in the blood that is converted to fibrin by thrombin during blood clotting. Fibrin monomers polymerise into protofibrils, which then coalesce to form a visible hydrogel when acquiring a critical length of $\sim 20$–25 fibrin monomers[55]. The AR of this critical length protofibril is calculated to be 44 on average[9,56] (see Supplementary Information). Figure 3d shows how the lag time of fibrin network formation varies with changing $\phi$ of fibrin in solution. The data looks remarkably similar in profile to the data obtained for $pL_7$ (Fig. 3b), suggesting that universal mechanisms may govern the formation of diverse protein networks. Similarly, fitting the curve and extracting $\phi_{crit}$ for fibrin networks yields a value of $(5 \pm 1) \times 10^{-2}\%$ (equivalent fibrin concentration of $0.7 \pm 0.1$ mg ml⁻¹). This value for $\phi_{crit}$ is within the error of $\phi_{crit}^{rod}$ (predicted with Eq. 1), demonstrating a transition from TDL to RDL formation is also present in naturally occurring networks. This transition is also observed in fibrin networks despite the stark difference in the non-covalent thrombin-mediated crosslinking mechanism of fibrin to the covalent photo-chemical crosslinking mechanism of pL polyprotein hydrogels. Note that the predicted value for the fibrin network comes from

considering fibrin protofibrils (AR ≈ 44), not the mature fibrin fibres or fibrin monomers, suggesting that the network formation is dominated by the formation of junction points[57] between protofibrils, not their bundling[55].

Importantly, the $\phi_{crit}$ value of fibrin networks coincides with $\phi$ values observed in patients diagnosed with hypofibrinogenemia[58,59] (shown as a shaded grey bar in Fig. 3d), i.e. low blood fibrinogen levels, which can result in improper clotting and bleeding. Fibrinogen blood level (concentration of fibrinogen in blood plasma) is of critical importance, as fibrinogen is the first coagulation protein to reach critically low concentrations in traumatic bleeding[60]. Consequently, clinical guidelines[61,62] recommend fibrinogen supplementation when the fibrinogen level falls below 1.5 mg ml⁻¹ in patients with traumatic bleeding. A recent study which supplemented fibrinogen-deficient plasma with increasing concentrations of fibrinogen found that 0.75 mg ml⁻¹ fibrinogen level was already sufficient to bring clotting time within normal range[63] (Note the authors also observed a similar profile in clot time vs. fibrin concentration as in Fig. 3d). Normal fibrinogen blood level (2–4 mg ml⁻¹)[62,63] are well above this region, suggesting that not only do biological systems exhibit these geometric rotational effects but exploit them for the rapid formation of isotropic homogeneous networks.

## Discussion

The geometric properties of a network building block play a key role in controlling the architecture and subsequent mechanics of hydrogel networks. By increasing the AR of protein building blocks through the inclusion of additional pL domains and comparing different AR proteins at a fixed protein volume fraction, we have demonstrated that the mechanical rigidity ($G'$) of the network increases with increasing AR up to a plateau value. A fixed and known protein volume fraction is crucial to allow for comparison of protein network formation across the different AR building blocks. This is distinct from previous studies, which have studied the mechanical properties of the swollen hydrogels after network formation and relaxation in a specific buffer of interest to exploit chain entanglement for creating cartilage-like material properties[64] and bi-layer swelling to engineer shape memory and morphing biomaterials[65].

Using a combination of rheology, control of building block geometry and course-grained simulations, we demonstrate that the increase in $G'$ is governed by two mechanisms; (i) the building block exceeds critical coordination when AR > 3 and (ii) the system becomes rotationally limited when $AR > AR_c$, as predicted by a rotating rod model. SANS results show that this rotational limit leads to an alteration in the topology of the pL hydrogel network from connected fractal-like clusters to a homogenous network of inter-connected building blocks, which leads to increased mechanical rigidity when $AR > AR_c$. Finally, an analysis of the lag time behaviour during network formation revealed a transition from TDL formation to RDL formation, in agreement with a rotating rod model. The results show that the transition from a TDL to RDL network formation causes a significant alteration to the network topology from connected fractal clusters to homogeneous networks, leading to increased mechanical rigidity. Interestingly, in addition to the effects of AR on hydrogel formation, AR has also been shown to affect the mechanical behaviour of collagen under deformation[66].

The results demonstrate that building block AR is a key design parameter in the formation, structure and mechanics of hierarchical protein networks. To demonstrate the universality of the effects we observe in our engineered hydrogels, we also investigated the lag behaviour of the naturally occurring network-forming protein fibrin. We show that fibrin also exhibits a transition from TDL to RDL network formation in accordance with a rotating rod model, suggesting that the geometric effects observed in synthetic protein hydrogel networks are universal to similar fibrillar networks found in biological systems.

Previous measurements by Morrow et al.[56] reported that the clotting time showed a reciprocal relationship with fibrin concentration (in agreement with our results) in the presence or absence of plasma and showed that the clotting time plateaued above fibrin concentrations of 0.75 mg ml$^{-1}$ (remarkably close to our extracted rotational crossover of $0.7 \pm 0.1$ mg ml$^{-1}$). From our work, we argue that this relationship between clot time and fibrin concentration is due to the crossover from TDL to RDL assembly and is an example of the advantage of high AR building blocks in structural networks, namely the consistently rapid formation of a network over a range of concentrations. This shows a distinct advantage of high AR protein building blocks over globular proteins (AR ~ 1), which our rotating model (Eq. 1) predicts would not exhibit a similar crossover in behaviour until a volume fraction of 78.5% (~1000 mg ml$^{-1}$) higher than it is theoretically possible to pack hard spheres[67,68].

This study suggests several advantages for high AR building blocks for network formation, including; (i) increased network rigidity at equal concentrations, maximising the impact of each protein produced on the network and thus minimising the amount needed to be produced; (ii) the formation of isotropic homogenous networks providing consistent architecture across the whole network and finally (iii) consistent formation time of the network over a wide range of building block concentrations which allow for a wide functional concentration range for crucial biological processes such as clot formation. Such advantages are likely to play an important role in a wide range of biological networks.

By understanding the crucial role of building block aspect ratio on the formation process of protein networks and subsequent effects on the network topology and mechanics, we demonstrate the importance of building block geometric properties on hierarchical networks. Biology exploits these geometric properties, offering the opportunity for materials science to do the same.

## Methods
### Materials
Tris(2,2′-bipyridyl)dichlororuthenium(II) hexahydrate (Ru(BiPy)$_3$), sodium persulfate (NaPS), sodium chloride (NaCl), calcium chloride (CaCl$_2$), tris(hydroxymethyl)aminomethane (TRIS), sodium phosphate dibasic and sodium phosphate monobasic were obtained from Sigma-Aldrich and used without further treatment. Similarly, fibrinogen (PDB code: 3GHG) and thrombin from human plasma were purchased from Sigma-Aldrich and used without any further treatment.

### Wild type protein L mutagenesis
The gene encoding protein L I11Y (pL) was made by Q5 mutagenesis of pseudo-wildtype pL (PDB code: 1HZ6 and was used as the monomeric pL subunit for all polyproteins. The assembly of genes encoding pL polyproteins was performed using a PCR-based Golden Gate protocol with a modified pET14b, with all original BsaI sites removed, as the destination expression vector. pL monomer, dimer (pL$_2$), trimer (pL$_3$), tetramer (pL$_4$), pentamer (pL$_5$), hexamer (pL$_6$) and heptamer (pL$_7$) were made using the Golden Gate protocol[27,28]. DNA sequences were then verified by Sanger sequencing (Source Bioscience Ltd., UK).

### Protein L poly-protein preparation
The pL polyprotein plasmids were transformed into BL21 (DE3) pLysS *E.coli* cells. 2 ml LB starter culture was used to inoculate 0.5 L auto-induction medium[69]. Totally, $10 \times 0.5$ L cell cultures were incubated at 28 °C, 200 rpm for 24 h, for protein expression and then harvested. Cell pellets were resuspended in lysis buffer (20 mM Tris, 300 mM NaCl, 10 mM Imidazole, 2 mM Benzamidine, 1 mM PMSF, 0.01% Triton-X-100, pH 8), passed through a cell disruptor at 30 kPsi, 4 °C and then centrifuged at 25,000 rpm, 25 mins, 4 °C. The supernatant containing overexpressed protein L constructs was loaded onto $2 \times 5$ ml HisTrap FF columns (Cytiva) equilibrated with wash buffer (20 mM Tris, 300 mM NaCl, 10 mM imidazole, pH 8). Unbound protein was eluted with wash buffer before pL was eluted with 100% elution buffer (20 mM Tris, 300 mM NaCl, 500 mM imidazole, pH 8). Protein was further purified using a 5 ml HiTrap Q HP anion exchange chromatography column (Cytiva) for ion exchange purification. A gradient elution of 0–100% elution buffer (25 mM sodium phosphate pH 7.4, 0–500 mM NaCl) was performed over 300 ml. Protein was finally purified by size exclusion chromatography (HiLoad 26/60 Superdex 75 pg, Cytiva) and eluted in 25 mM sodium phosphate, pH 7.4. The purified protein was dialysed into ultra-pure water (Milli-Q, resistivity ≥18.2 MΩ cm) and lyophilised for storage at −20 °C.

### Sample preparation and gelation
**pL hydrogels.** Hydrogel samples were prepared by mixing a concentrated crosslinking reagent stock (2× or 4× of final sample final concentration: 30 mM NaPS, 100 μM Ru(BiPy)$_3$, 25 mM sodium phosphate, pH 7.4) and a concentrated protein stock (67 mg ml$^{-1}$ or 50 mg ml$^{-1}$ in 25 mM sodium phosphate, pH 7.4), in either a 1:3 or 1:1 ratio. The concentration of protein stocks was confirmed by measuring Abs$_{280}$ of 100× dilutions using the molar extinction coefficients and molecular masses of each protein L construct (Supplementary Information). Photo-chemical gelation was initiated and controlled using a blue LED (peak emission at 450 nm) at a current of 0.48 Amps. The photochemical crosslinking method used in this work was devised by Fancy et al. in 1999 and is mediated by Ru(BiPy)$_3$ causing the formation of tyrosine free radicals, which then react with other tyrosines to form dityrosine bonds[70].

### In situ formation of fibrin networks
Measurement of the lag time and the initial phase of fibrin gelation was conducted in situ on the rheometer. 900 μL fibrinogen solution (ranging from 0.11 mg ml$^{-1}$ to 2.4 mg ml$^{-1}$) was loaded onto the rheometer and at $t = 60$ s, 100 μL activation mix (10 U ml$^{-1}$ thrombin and 50 mM calcium chloride) was added directly. The final concentrations in the reaction mixtures were 1 U ml$^{-1}$ thrombin and 5 mM calcium chloride. Thrombin was added to calcium immediately before mixing the activation mix with the fibrinogen solution to ensure maximum activity. All solutions and mixtures were made in TBS buffer (50 mM Tris, 100 mM NaCl, pH 7.4).

### Rheometry
Mechanical characterisation of pL hydrogels and fibrin networks was performed on an Anton Paar MCR 502 stress-controlled rheometer (Anton Parr GmbH, Austria) in pseudo-strain-control mode, using. A parallel plate configuration was used for both pL (8 mm diameter) and fibrin (25 mm diameter). Time sweep gelation measurements on both pL hydrogels and fibrin networks were conducted at a frequency and shear strain of 1 Hz and 0.5%, respectively, while frequency sweep measurements of pL hydrogels were conducted at a shear strain of 0.5%. To prevent evaporation and water uptake, during the rheology measurements, low-viscosity silicone oil (approximately 5 ct) was placed around the sample and geometry. Storage moduli extracted from frequency sweeps at 1 Hz (Fig. 1b) were fitted with a dual sigmoid function;

$$G'(\text{AR}) = G'() \cdot \left( \frac{\alpha}{1 + e^{-k_z(\text{AR} - \text{AR}_z)}} + \frac{1 - \alpha}{1 + e^{-k_c(\text{AR} - \text{AR}_c)}} \right) \quad (4)$$

where $G'(\infty)$ is the fitted storage modulus at an infinite building block aspect ratio; $\alpha$ is the proportion of $G'(\infty)$ due to the network building blocks surpassing critical coordination; $A_z$ and $A_c$ are the aspect ratios when the network building blocks pass critical coordination (Fig. 1c) and rotational limit (Eq. 1d, Eq. 1), respectively; finally $k_z$ and $k_c$ are related to the width of the transitions centred around AR$_z$ and AR$_c$, respectively, at which $G'$ increases to plateau when the system passes

critical coordination and rotational limit, respectively. Rheology analysis and figures were produced in OriginPro 2021.

### Determination of network lag time

The lag time of network formation of both pL hydrogels and fibrinogen was determined from rheological gelation curves (time sweeps) at a constant frequency of 1 Hz and constant strain of 0.5%. To extract the lag time, the gelation curves of $G'$ vs. gelation time were fitted with two linear lines. The first line was fitted to the pre-gelation G' values, while the second was fitted to the maximum slope of the polymerisation curve (i.e. where $G'$ is increasing linearly with time). The lag time is then determined from the intersection of these two fitted linear lines. Graphical examples of these extractions can be seen in Supplementary Fig. S13a, b.

### Small angle neutron scattering (SANS)

SANS measurements were conducted on the time-of-flight instrument Sans2d at the ISIS Neutron and Muon Source (STFC Rutherford Appleton Laboratory, Didcot, UK). Sans2d front and rear detectors were set up at 5 and 12 m, respectively, from the sample, defining the accessible $q$-range as 0.0015–1 Å$^{-1}$. Temperatures were controlled by an external circulating thermal bath. Samples were loaded and gelled in 1 mm path-length quartz cuvettes. To prevent the exchange of deuterium with hydrogen in the air, evaporation, and water uptake, the quartz cuvettes were sealed with parafilm. The pL hydrogel samples were made and loaded in pairs into the Sans2d autosampler before being measured, e.g. pL$_7$ hydrogels at protein concentrations of 25 mg ml$^{-1}$ and 50 mg ml$^{-1}$. The accumulation time of pL hydrogel samples varied by protein concentration, where 25 mg ml$^{-1}$ samples were measured for approx. 30 min (20 μAmp h) and 50 mg ml$^{-1}$ samples were measured for approx. 15 min (10 μAmp h). The raw SANS data were processed using the Mantid framework following the standard procedures for the instrument (detector efficiencies, measured sample transmissions, absolute scale using the scattering from a standard polymer, etc.).

SANS curves were fitted using SASview (http://www.sasview.org) in accordance with Eq. 5.

$$I(q) = \phi \Delta \rho^2 V_{pL} \cdot \left[ (1 - P_c) \cdot F_{pL}(q) + \frac{3}{2} \cdot AR \cdot P_c \cdot F_{rod}(q) \cdot S(q) \right] + \cdot bkg$$

(5)

Where $F_{pL}(q)$ is a spherical form factor modelling the scattering of a single pL domain in the polyprotein chain, and $F_{rod}(q)$ is a cylindrical form factor modelling the scattering of the rod-like pL building block (the factor of 3/2 is the identity between a cylinder and a sphere, i.e. at AR = 1, $V_{rod} = 1.5V_{pL}$). Finally, $S(q)$ is a fractal structure factor[71,72].

$$S(q) = \frac{D_f \Gamma(D_f - 1)}{\left[1 + \frac{1}{(q\xi)^2}\right]^{\frac{D_f - 1}{2}}} \cdot \frac{\sin\left[\left(D_f - 1\right)(q\xi)\right]}{(qR_0)^{D_f}}$$

(6)

From the fractal structure factor (Eq. 6), the fractal dimension, $D_f$, and correlation length, $\xi$, can be extracted. $R_0$ is the effective spherical radius of the network building block.

### Conversion from concentration to volume fraction

Throughout this work, the concentration of both protein L and fibrinogen is converted into volume fraction and vice versa. Equation 7 shows the formula for the conversion from protein concentration in mg ml$^{-1}$ to protein volume fraction,

$$\phi = \frac{conc\left(\frac{mg}{ml}\right)}{1000 \cdot \rho_{prot}\left(\frac{g}{cm^3}\right)}$$

(7)

where $\rho_{prot}$ is the average density of protein (which in this work is taken to be 1.37 g cm$^{-3}$)[73,74]. The water content of hydrogels, $\phi^{water}$, is calculated by taking the protein volume and subtracting it from unity, $\phi^{water} = 1 - \phi$. Note that the protein concentration in mg ml$^{-1}$ is determined via the absorption at 280 nm in conjunction with the Beer–Lambert Law.

### Computational modelling

In this work, two computational models were employed to model both the average coordination of each building block (BioNet) and extract the fractal dimension of the predicted structure as a function of AR without the rotation of the building block included (kinetic lattice model).

### BioNet

BioNet is a dynamic model which utilises Brownian dynamics to simulate the diffusion and interaction of arbitrary objects at biological length scales. BioNet also implements a kinetic bond formation protocol, allowing the dynamic objects to form Hookean springs dynamically over the course of a simulation. Uniquely, BioNet explicitly models (cross-linking) binding sites as points explicitly at the surface of the moving objects. Together with a steric repulsion and rotational degrees of freedom included, this allows network percolation and rigidity to emerge naturally as the result of the geometry and topology of the dynamic subunits without the need for complex angular interaction potentials. Importantly for this work, BioNet has been used previously to study the emergence of persistence length in single oligomeric structures[36] and the formation of protein-based hydrogel networks from collections of these oligomeric subunits[38].

In the present work, each protein L representative subunit was modelled as a spherical subunit of radius, $R = 15$ Å. These subunits were given four equally distributed binding sites and subsequently assembled into oligomers with AR values in the range of those used experimentally. Cubic boxes with periodic boundary conditions were populated with $N = 500$ oligomers. The size of these boxes was calculated such that the material volume fraction was that used experimentally. These systems were then thermodynamically equilibrated before kinetic bond formation was activated, simulating the hydrogel systems being exposed to light. The final states of the simulated networks were then analysed for their average coordination properties, as seen in Fig. 1c.

### Kinetic lattice model

The coarse-grained lattice model simulated the aggregation and translational diffusion of clusters of monomeric units, where each unit represented a single protein L. Rod-like polymers, with geometry AR units in length and one unit in width, were sequentially added to a cubic periodic simulation cell, with random position and orientation along a lattice vector and the constraint that no two could overlap until the target volume fraction was reached. During a simulation run, all polymers were randomly diffused and reacted (cross-linked) when the surfaces of two different clusters became adjacent, forming increasingly extended polymer clusters, through gelation, to a final network in which every monomeric unit belonged to a single cluster. The box covering the distribution of the final state, including fractional counts for improved fitting[75], was measured, and the fractal dimension $D_f$ was extracted using a fitting function that included the expected crossover to homogeneity[76], removing the need to visually identify power-law regions. For this work, simulations were run at monomeric volume fractions of 1.85 and 3.7% and building block aspect ratios AR ranging

between 3 and 7, with 10 repeats for each combination. Further details can be found in previous work by Cook et al.[76].

## Reporting summary

Further information on research design is available in the Nature Portfolio Reporting Summary linked to this article.

## Data availability

Source data is provided with this paper in the Aspect Ratio Controls Protein Networks Repository, found at: https://doi.org/10.5518/1344. All other data are available from the corresponding author upon request.

## Code availability

Access to the code repositories is found at https://doi.org/10.5518/1344.

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

## Acknowledgements

The project was supported by a grant from the Engineering and Physical Sciences Research Council (EPSRC) (EP/ P02288X/1) and a European Research Council Consolidator Fellowship/ UKRI Frontier Research Fellowship for the MESONET project UKRI EP/X023524/1 to L. Dougan. We acknowledge support from the EPSRC Multiuser equipment grant EP/ V035460/1. We acknowledge ISIS Neutron and Muon Source for access to the Sans2d beamline (experiment number RB2210126, https://doi.org/ 10.5286/ISIS.E.RB2210126). This work benefitted from SasView software, originally developed by the DANSE project under NSF award DMR-0520547. Additionally, we acknowledge funding from the Wellcome Trust for Chirascan, grant code 094232, and G. Nasir Khan for his support. We would like to thank Adrian Cuncliffe for his help and mechanical support with thermogravimetric analysis. Work in the Ariens lab is supported by the British Heart Foundation (RG/18/11/34036), Wellcome Trust and (204951/B/16/Z) the BBSRC (BB/W000237/1). Many thanks to all members of the Dougan group for helpful discussion and feedback.

## Author contributions

Experiments were conceived and designed by M.D.G.H., S.C., D.J.B. and L.D. Protein L constructs were designed by M.D.G.H. and S.C. Protein L polyprotein preparation (including molecular biology, expression and characterisation) was completed by S.C. Fibrinogen samples were provided and prepared by T.F and R.A. Rheology data collection, analysis and theory was done by M.D.G.H. SANS and SAXS data was collected by M.D.G.H, S.C. and N.M. SANS and SAXS analysis was done by M.D.G.H.

CD data collection and analysis was performed by M.D.G.H. and S.C. DLS data collection and analysis was performed by D.L.B. Computational modelling was done by B.S.H., K.R.C. and D.A.H. Figure design, paper writing and editing was done by M.D.G.H., S.C., B.S.H., K.R.C., T.F., N.M., D.L.B., J.M., R.A., D.A.H., D.J.B. and L.D. Project supervision and funding were provided by D.A.H., D.J.B. and L.D.

## Competing interests

The authors declare no competing interests.
