## [Peer Review File · Nature Communications]

Biology Exploits Geometry: Impact of Aspect Ratio on Protein NetworksREVIEWER COMMENTS

Reviewer #1 (Remarks to the Author):

The authors report high aspect ratio (AR) fibrous networks based photocrosslinked gel study constructed from protein building blocks with varying numbers of protein L (pL) domains with ARs from 1-7 and characterised them using shear rheology and small angle neutron scattering (SANS) and computational simulation. I have following comments/action for the work.

While authors describe that AR is a crucial property that defines network architecture and mechanics, however the gel structures are studied as such as hydrogel (Hydrogels constructed from each building block were analysed by a multiscale experimental approach including rheology and small angle neutron scattering (SANS) to observe the changes in the mechanics and network topology as AR was varied. Interestingly, there is no data on water uptake of the studied samples in rheology, SANS (where probably D2O has been used). The authors studied two distinct network types with AR 3 and 7 and also at different concentrations. But the discussion does not reflect the mesh size in each case. SANS experimental protocol needs to be highlighted in detail and also highlight the SLD values and also the data show upturn at low Q region, the authors might consider to go to another magnitude lower Q range or even the USANS range.

The authors should highlight also on the crosslinking mechanism, as it induced by Ru (biPy)₃. What is the tyrosine content of the protein studied for crosslinking?

Could the authors clarify what is the role of water on the networks with increased AR building blocks to exhibit more homogeneous structures and higher storage moduli and how it affects the shift from translational diffusion limited (TDL) to rotationally diffusion limited (RDL) network formation and the lack of space to rotate freely. As a comparative study the authors also investigated the fibrin network and observe the same transition from TDL to RDL formation, confirming that living systems exploit AR for their network assembly. But in case of fibrin the phenomenon occurs under a different condition. Also the elasticity of the AR network is important for the overall mechanics. What is the elasticity level of these AR network based pL gels. What are the sizes of the homogeneous and connected cluster size with change in AR values in the system.

The authors need to confirm how the network formation kinetic was studied and how with different water content the rigidity changed and also whether any glass transition behavior noted.

Although the protein was obtained from low to higher mol wt ranges, ultimately pL 7 was studied extensively. The authors also mention that free rotation limit in hydrogel network formation is explained with respect to lag time, which is determined both experimentally from rheology and simulation and beyond certain volume % the transition from TDL to RDL assembly. However, in absence of any structural information or CD spectroscopy data, it is not clear what is driving force for the fiber network formation in pL protein. Can the author provide data on CD spectroscopy on the protein structure? This might provide clue on how to manipulate AR

Overall, the paper describes in-depth physics of the AR driven network formation, but structural and chemical information would help to make the data more meaningful.

Reviewer #2 (Remarks to the Author):

This manuscript investigates how the aspect ratio of protein building blocks may change morphology and mechanics in hydrogel networks. The authors combine rheology and scattering experiments with a numerical simulations platform specifically designed to study protein hydrogels. The results obtained suggest that higher aspect ratio (AR) of the protein building blocks may lead to more spatially homogeneous networks, which can also be mechanically stiffer. This hypothesis is tested in a model protein hydrogel and in natural (fibrin) protein gels, which are central to blood clotting. Based on the results obtained, the authors propose that the AR of protein building blocks should be considered as a design principle, developed through evolution, in biologically relevant materials.

The paper is certainly interesting and the combination of different approaches, and of different systems, is valuable and constitute a significant contribution. However, I find that a few points need clarification and, potentially, integrating additional analysis and/or justification.

1) The change in the network topology and architecture are mainly demonstrated, as far as I understand, using the "fractal dimension" D_f extracted from the scattering data (example in Fig. 21, but several other dataset are provided in the paper and SI). I find this debatable, since the power law regime in the scattering intensity curves is hardly detectable, and, even so, very limited in range. While it may be possible to fit the scattering data with many parameters (see Methods), one of which being D_f , covering various regimes and transients between them, this approach cannot provide a strong enough evidence of a fractal regime. Furthermore, what the authors indicate as fractal aggregates, with fractal dimension changes correlated to the different translational/rotational diffusion regimes, are aggregates roughly of the same linear size of the building blocks (rods) themselves, as suggested also in the cartoon of Fig. 2(e), so I don't see how they could be fractal objects.

In my view, the changes in the network morphology and topology would be better demonstrated by showing changes in the q -ranges of the different regimes with protein concentration and AR, potentially providing evidence of the changes in the mesh-sizes of the networks with respect to the size of the aggregates.

2) One of the main ideas discussed in the paper is that the AR of the protein building blocks changes the prevalence of translational vs rotational diffusion and that this mechanism significantly modifies the network architecture, with consequences for morphology and mechanics. In Figure 1 the authors provide indirect evidence that this mechanism may be the relevant one by fitting the dependence of the gel elastic moduli on AR to obtain a critical aspect ratio at which the transition from prevalently translational to rotational diffusion would occur, and using the average coordination of the protein units in the networks built through the numerical simulations. For the fibrin gels, instead, they interpret the network formation lagtime as an indication that the same mechanism is present. However, it should be possible to measure this transition in the diffusion regimes using quasi-elastic scattering, to extract diffusion constant or demonstrate two distinct regime in the dynamics of the rods as the network forms, correlated to the distinct regimes detected in $G'(AR)$. This information should be accessible to both

experiments and/or simulations.

3) The discussion on the average coordination also raises some doubts, especially since these networks are prevalently very heterogeneous (as discussed in various parts of the manuscript) hence the average coordination cannot provide much information about rigidity. Have the authors performed rheological tests in the simulations? Or have they the possibility to establish that those are rigid in some other way?

Also the relevance of rigidity criteria valid for athermal systems with frictional forces seems debatable for the protein gels. The authors may want to consider rather the literature on biopolymer and fiber networks, for example some of the works by the Koenderink group, showing how the network rigidity and mechanics may be controlled by both the geometry of the network branching and by the larger scale heterogeneities in a broad range of biopolymer networks (see for example Broedersz et al, Nature Phys. 2011 for rigidity, or Burla et al. PNAS 2020, for the network failure).

Reviewer #3 (Remarks to the Author):

In this manuscript, Hughes et al. perform an experimental and computational study to investigate the role of aspect ratio (AR) of protein building blocks in the formation of hierarchical network structure. They report that network homogeneity and mechanics depend on AR, and ascribe the change from a heterogenous, weaker network to a homogenous, stiffer network to an underlying change in dynamical degrees of freedom of the constituent building blocks.

This is a well motivated study that takes a novel (to my knowledge) approach to studying network formation. I was interested to read about the findings and learn more about the mechanism. The experimental data appear sound, and the BioNet modelling provides additional insight to support the conclusions. Given the presence of high-AR protein building blocks in nature (e.g. fibrin and collagen), and in using high-AR structures for connected active matter network systems (e.g. actin and microtubules), I think there is likely to be considerable interest in this work.

My concerns with the work lie with the robustness of conclusions gleaned from experimental data. For example, the experimental work using protein L is predicated upon the assumption that pL polyproteins with 4-aa linkers act as stiff rod-like objects. However, there is no evidence presented in the manuscript to demonstrate that this is the case. It could be possible, for example, that a 4-aa linker is sufficiently short to introduce a kink between neighbouring pL domains, so that instead of poly-pL being a straight rod, it is a bent/kinked rod. Can the authors demonstrate that their poly-pL constructs indeed are structured as straight rods? If not, can they comment on the implications of distinct structures such as a curved rod, in the context of their model?

Furthermore, the structure of pL included in Fig. 1a seems to already have an AR greater than 1: perhaps closer to 1.5 or 2. In the clearly described BioNet simulations, pL is modelled as two connected spheres,

implying an AR close to 2. Why is the AR of a single pL assumed to be 1 throughout the presentation of the experimental work? How much does this assumption affect the interpretation of results?

The agreement between equation (1) and the data for ARc vs concentration is striking (Figure 1d) and seems to be strong evidence in favour of the model of rotational diffusion contributing to the transition behaviour of G' . However, it is not clear how the authors have converted from concentration to phi, volume fraction. Is the conversion factor experimentally determined or found via this fit? The conversion between phi and concentration seems to be a key parameter that can support the conclusions of this work, and it would be good to show whether this value is physically reasonable. From my back-of-the-envelope estimates, it appears that the phi crit value should be closer to 3.8% if ARcrit is 4.52, in contrast to the value of 1.6% provided following equation (3). It would also be good to provide the range of concentrations / phi values for which their experimental results are in the dilute regime (discussion following equation (2)). The discussion around phi and concentrations and the different regimes of dynamics for fibrin would also be assisted by providing the reader a sense of how these values are converted.

The authors also provide simulations from a kinetic lattice model as support for their rotational diffusion mechanism. The model does not include rotation and the results of the simulations show a linear, rather than sigmoidal, trend in Df. Disagreement between these results and the experimental findings does not “confirm[ing] that rotational diffusion is a key mechanism driving the architecture of protein networks”. The model is unable to confirm this, without explicitly adding rotational diffusion and showing that the sigmoidal trend in Df is recovered. I do not find that the kinetic lattice model is useful in helping to provide a mechanistic understanding of network assembly and structure. The cited reference for more details about the model (ref 64) is also not useful at the current time.

The storage modulus as a function of AR is clearly not well described as a sigmoid. However, the use of a double sigmoid function is concerning, given the number of fit parameters (6) and data points (7). With so many degrees of freedom, it should be easy to fit their data to this function. At a minimum, the authors should report statistical goodness of fit parameters such as reduced chi squared.

Minor points:

What do the error bars in Figure 1b represent, and how many replicates were performed for each measurement?

What aspect ratio are the networks in the insets to Figure 1c meant to depict?

It is unclear what the star-like objects in the schematic insets of Figure 1d are meant to represent.

I am not sure what the importance/relevance of the following sentence is. I suggest clarifying or eliminating, to avoid confusion. “Note: 2D thermal simulations have been performed which

demonstrated that a significant increase in the modulus was still observed at $\zeta = \zeta_c$ despite the inclusion of thermal fluctuations to the simulated network.”

Why are the data points for D_f vs AR in Fig 2d fit to a sigmoid model only for 50 mg/ml? It looks like a sigmoid could also describe the 25 mg/ml data.

“It is intuitive that higher AR building blocks will link together to form much more porous structures”. My intuition could tell me the opposite: that higher AR building blocks can pack laterally, as is the case in collagen fibrils, and make structures whose backbones are highly dense and aligned. Can this statement in the manuscript be clarified?

It was only in the methods section that I realized that gel formation was triggered by light exposure. It could be helpful to add this earlier in the manuscript.

All parameters from fits to data should be provided for the reader (e.g. in a table in the SI). These include all six parameters from each double-sigmoid fit.

It is not clear to me how rates influence G' as measured in Figure 1b. It is also not clear why this rate * AR gives a unitless number. Please clarify the basis for the form of equation (4).

The loss ratio is not defined in the text.

The abstract states that network formation is more rapid with high AR. I may have missed it, but I do not see this explicitly addressed in the text. When I look at Figure 3, I see that pL provides a much shorter lag time than fibrin, although it has a shorter AR. Please check this.

28th April 2023

Dear Reviewers

Thank you for the reviews of our manuscript.

Yours faithfully,
Professor Lorna Dougan

Please find below the reviewers comments in blue and our response in black.

Reviewer #1

Q1.1 Interestingly, there is no data on water uptake of the studied samples in rheology, SANS (where probably D2O has been used).

Water uptake was not the focus of this study and rheology and SANS experiments were conducted under conditions which minimise evaporation. From the rheology data we can observe the normal force of the sample during gelation which is indicative of swelling or deswelling of the network. The data indicates minimal deswelling occurs, as expected as evaporation is minimised in our rheology experiments through the use of a silicon oil sealing method at the gel-air interface.

We have included text to comment on the normal force and have included figure S13 in the supporting info.

(Page 7) *“During gelation we note that there is a small increase in the normal force in the negative direction (Fig. S13) suggesting that the gels are undergoing slight deswelling.”*

Q1.2 The authors studied two distinct network types with AR 3 and 7 and also at different concentrations. But the discussion does not reflect the mesh size in each case. SANS experimental protocol needs to be highlighted in detail and also highlight the SLD values and also the data show upturn at low Q region, the authors might consider going to another magnitude lower Q range or even the USANS range.

We have included additional details in the methods section regarding the experimental protocol.

(Page 11) *“Sans2d front and rear detectors were set up at 5 and 12 m, respectively, from the sample, defining the accessible q-range as 0.0015– 1 Å⁻¹. Temperatures were controlled by an external circulating thermal bath. Samples were loaded and gelled in 1 mm path length quartz cuvettes. The pL hydrogel samples were made and loaded in pairs in to SANS2D autosampler before being measured e.g. pL7- hydrogels at protein concentration of 25 mg/ml and 50mg/ml. The accumulation time of pL hydrogel samples varied by protein concentration where 25mg/ml samples were measured for approx. 30 minutes (20 μAmp·hr) and 50 mg/ml samples were measured for approx.. 15 minutes (10 μAmp·hr) The raw SANS data were processed using the Mantid framework following the standard procedures for the instrument (detector efficiencies, measured sample transmissions, absolute scale using the scattering from a standard polymer, etc.).”*

Additionally, we include the extracted correlation length (which are on the order of 100s Å) in the supplementary information (Figure S7), which demonstrates that the scale of the network structure is within the Q-range accessible to SANS2D (Q-range: $0.0015 \text{ \AA}^{-1} - 1 \text{ \AA}^{-1}$; equivalent length scales in d-space: $4200 \text{ \AA} - 6.3 \text{ \AA}$).

Table S1 provided in supplementary information containing the SLDs in 100% D₂O for each protein L construct.

The authors should highlight also on the crosslinking mechanism, as it induced by Ru (biPy)₃. What is the tyrosine content of the protein studied for crosslinking?

We have added and modified text to clearly state the number of tyrosines per protein L domain.

(Page 2) *“The variant used in this study has 4 surface exposed tyrosine residues (a critical coordination of 4 is required to form a continuous self-supported network in an athermal frictional system^{35,36}).”*

Additionally, we have added text that briefly describes the crosslinking reaction in the methods and signposts to Fancy et al. 1999 paper in which the method was devised.

(Page 11) *“The photochemical crosslinking method used in this work was devised by Fancy et al. in 1999 is mediated by Ru(BiPy)₃ causing the formation of tyrosine free radicals which then react with other tyrosines to form dityrosine bonds⁷¹.”*

Q1.3 Could the authors clarify what is the role of water on the networks with increased AR building blocks to exhibit more homogeneous structures and higher storage moduli and how it affects the shift from translational diffusion limited (TDL) to rotationally diffusion limited (RDL) network formation and the lack of space to rotate freely.

A changing solvent viscosity would impact the translational and rotational diffusion of the protein building blocks. However, the role of water on AR network formation is not the focus of this study and the solvent environment was unchanged through the experiments. Instead our focus is on the AR/geometry of the protein building block, and Eqn 1 models the transition from TDL to RDL (also known as the transition from dilute to semi-dilute regime for rod-like particles) without the need to include the solvent.

As a comparative study the authors also investigated the fibrin network and observe the same transition from TDL to RDL formation, confirming that living systems exploit AR for their network assembly. But in case of fibrin the phenomenon occurs under a different condition. Fibrin network formation occurs via different crosslinking method, i.e. thrombin mediated non-covalent bonds, to the covalently photo-chemically crosslinked pL poly proteins. Despite this difference in crosslinking mechanism, we observed that both system exhibit the effect that is accurately modelled by our geometric model (Eqn. 1).

We have included additional text to highlight this point

(Page 8-9) *“This transition is also observed in fibrin networks despite the stark difference in the non-covalent thrombin mediated crosslinking mechanism of fibrin to the covalent photo-chemical crosslinking mechanism of pL polyprotein hydrogels.”*

Q1.4 Also, the elasticity of the AR network is important for the overall mechanics. What is the elasticity level of these AR network based pL gels?

To comments on the elasticity level of the AR network based pL gels, we have provided $\tan(\delta)$ values in the supplementary (Figure S4) and have added text to the main manuscript to comment on this elasticity.

(Page 3) *“This is confirmed by the loss ratio (defined as a ratio loss modulus to the storage modulus) (Fig. S4), which is above one for low AR samples ($AR < 3$) exhibiting fluid dominated behaviour as opposed to the viscoelastic solid-like behaviour expected for a gel. For gels constructed from building blocks of AR three and greater, we observe loss ratios of approx. 0.03 demonstrating that these gels are self-supporting networks which are dominated by their elastic behaviour.”*

Q1.5 What are the sizes of the homogeneous and connected cluster size with change in AR values in the system?

From the scattering model we use to fit our SANS data we can extract key structural information such as the fractal dimension, D_f , and the characteristic length, ξ . The correlation length is a measure of the characteristic size of the mesoscale network structure.

We have added a graph of the characteristic lengths (Figure S7) in the supporting information and added text to the manuscript.

(Page 6) *“...dimension of the network, D_f , and the characteristic length of the network structure, ξ .”*

(Page 6) *“Interestingly, a similar trend is not observed in ξ as a function of AR, instead samples at 50mg/ml show no significant change in ξ ($\sim 280 \text{ \AA}$) as AR is increased while those at 25mg/ml exhibit a linear decrease from $\sim 700 \text{ \AA}$ at $AR=3$ to $\sim 500 \text{ \AA}$ at $AR=5$ ”*

Q1.6 The authors need to confirm how the network formation kinetic was studied and how with different water content the rigidity changed and also whether any glass transition behaviour noted.

We have added to the main text and the materials and methods to discuss how the lag time of the pL hydrogels and fibrin networks was measured. Additionally, we have signposted more clearly to figure S12 in the supporting information which shows exemplar gelation curves of both pL₇ and fibrin networks, highlighting how the fitting and extract of the lag time is performed.

Updated text on page 11

“Determination of Network Lag Time

The lag time of network formation of both pL hydrogels and fibrinogen was determined from rheological gelation curves (time sweeps) at a constant frequency of 1Hz and constant strain of 0.5%. To extract the lag time the gelation curves of G' vs gelation time were fitted with two linear lines. The first line was fitted to the pre-gelation G' values, while the second was fitted to the maximum slope of the polymerisation curve (i.e. where G' is increasing linearly with time). The lag time is then determined from the intersect of these two fitted linear lines. Graphical examples of these extractions can be seen in figures S12a and b.”

As stated above water content was not the focus of this study and as such we do not study how the rigidity is affected by alter water content. However, we do investigate how altering the protein concentration alters the hydrogel network. We have added text to discuss how increasing the

protein concentration alters the mechanical rigidity of the system. Furthermore, we highlight how this shifts the rotational limit of the system more clearly.

(Page 3) *"...further. Additionally, we observe that as the concentration of protein increases, we observe an enhanced rigidity of the networks that are formed, likely due to the additional load-bearing material present in the system. The..."*

(Page 5) *"...against protein concentration (Eqn. 1), demonstrating that by increasing the protein concentration the critical aspect ratio for rotational limit is decreased. This..."*

We do not observe any glass transition in our pL hydrogels as the volume fraction of the pL hydrogels is between 0 - 4% which is an order of magnitude lower than the approx. 30% needed for colloidal glasses, for example see figure 3 in Poon, MRS Bulletin, 2004, 29(2).

Q1.6 Although the protein was obtained from low to higher mol wt ranges, ultimately pL 7 was studied extensively. The authors also mention that free rotation limit in hydrogel network formation is explained with respect to lag time, which is determined both experimentally from rheology and simulation and beyond certain volume % the transition from TDL to RDL assembly. However, in absence of any structural information or CD spectroscopy data, it is not clear what is the driving force for the fiber network formation in pL protein. Can the author provide data on CD spectroscopy on the protein structure? This might provide clue on how to manipulate AR.

Protein L polyprotein rods are formed via protein engineering and recombinant growth in *E.coli*. Our structural SANS data shows that fibers (similar to those in fibrin networks) are not observed in pL networks. The driving force of network formation in our protein L hydrogels is the tyrosine specific photo-chemical crosslinking method. Despite this difference in crosslinking mechanisms and the lack of fibrous structures in pL relative to fibrin, we still observe similar agreement in the lag time behaviour of these network vs volume fraction, and both appear to exhibit a transition from TDL to RDL formation as predicted by our geometric model (Eqn 1).

Additional CD data for all pL AR constructs is provided in the supporting information (Figure S1), confirming that the protein L domains are in a native folded state in each AR polyprotein construct.

(Page 2-3) *"To confirm our pL polyprotein constructs were in a folded state we performed circular dichroism (CD) spectroscopy measurements on our polyprotein constructs. Figure S1 shows that the spectra for all the pL polyproteins have the same profile as the pL monomer, demonstrating that the pL polyproteins are in a folded confirmation, thus allowing us to control the aspect ratio by controlling the number of tandem repeats."*

Reviewer #2

Q2.1 The change in the network topology and architecture are mainly demonstrated, as far as I understand, using the "fractal dimension" D_f extracted from the scattering data (example in Fig. 21, but several other datasets are provided in the paper and SI). I find this debatable, since the power law regime in the scattering intensity curves is hardly detectable, and, even so, very limited in range. While it may be possible to fit the scattering data with many parameters (see Methods), one of which being D_f , covering various regimes and transients between them, this approach cannot provide a strong enough evidence of a fractal regime.

Our previous studies of folded protein hydrogels have identified that photo-chemically crosslinked folded protein hydrogels have fractal-like structures present (Hughes et al., *Soft Matter*, 2020, 27; Hughes et al., *ACS Nano*, 2021, 15(7); Aufderhorst-Roberts et al., *Biomacro.*, 2021, 21(10); Hughes et al., *ACS Nano*, 2022, 16(7)). Additionally, scanning electron microscopy (SEM) results on BSA hydrogels from the Popa group show very porous fractal-like structures. See figure 1e in Khoury et al., *Nat Comms.*, 2019, 10. We hence use this model to extract relevant structural parameters.

We also perform model independent Guiner-Porod fits to extract the Porod exponent as a function of AR at 25 and 50 mg/ml (Figure S9).

We have added additional text and an additional figure to the supplementary info (Figure S9).

(Page 6) *“Previous characterisation of folded protein hydrogels has identified fractal-like structures present in the network architecture^{26,45-47}.”*

(Page 6) *“To confirm these trends observed in fractal dimension are not a result of the model chosen, we also perform model independent Guiner-Porod fits to the low/mid-q region in order to extract the Porod exponent (i.e. the power law exponent). Figure S9 shows similar trends in the Porod exponent as a function of AR that are exhibited by the fractal dimension values in figure 2d.”*

Q2.2 Furthermore, what the authors indicate as fractal aggregates, with fractal dimension changes correlated to the different translational/rotational diffusion regimes, are aggregates roughly of the same linear size of the building blocks (rods) themselves, as suggested also in the cartoon of Fig. 2(e), so I don't see how they could be fractal objects.

We have added the characteristic length scale of the fractal clusters in the supplementary information (figure S7), demonstrating that the size of the clusters are larger than individual building blocks. However, in order to avoid reliance solely on a fractal model we have performed Porod analysis as discussed above, which demonstrated similar trends to those observed in the fractal dimension.

Q2.3 In my view, the changes in the network morphology and topology would be better demonstrated by showing changes in the q-ranges of the different regimes with protein concentration and AR, potentially providing evidence of the changes in the mesh-sizes of the networks with respect to the size of the aggregates.

The Guiner-Porod fits and extracted Porod exponent exhibits a clear transition at 50mg/ml (Fig. S9) which agrees with that observed in our fractal dimension plot (Figure 2d).

Q2.4 One of the main ideas discussed in the paper is that the AR of the protein building blocks changes the prevalence of translational vs rotational diffusion and that this mechanism significantly modifies the network architecture, with consequences for morphology and mechanics. In Figure 1 the authors provide indirect evidence that this mechanism may be the relevant one by fitting the dependence of the gel elastic moduli on AR to obtain a critical aspect ratio at which the transition from prevalently translational to rotational diffusion would occur, and using the average coordination of the protein units in the networks built through the numerical simulations. For the fibrin gels, instead, they interpret the network formation lagtime as an indication that the same mechanism is present. However, it should be possible to measure this transition in the diffusion regimes using quasi-elastic scattering, to extract diffusion constant or demonstrate two distinct regime in the dynamics of the rods as the network forms, correlated to

the distinct regimes detected in $G'(AR)$. This information should be accessible to both experiments and/or simulations.

We have performed additional DLS experiments on protein L 7-mer at the same concentrations as our lag time measurements. From these experiments we can extract the translational diffusion constant as a function of pL_7 volume fraction. These results show a sigmoidal transition centred at $2 \pm 0.2 \%$ which is close to the value predicted from our geometric model (Eqn 1) 1.6%.

These DLS results suggest that as the concentration of protein is increased above a critical concentration the system shift from a dilute to a semi-dilute regime, which would result in a shift from TDL to RDL formation.

We have added additional text to the manuscript discussing this as well as additional graphs (figure S14) to the supplementary information.

(Page 8) *“To confirm that there is a transition from the dilute (i.e. TDL) to the semi-dilute (i.e RDL) regime in pL_7 solutions, we perform dynamic light scattering (DLS) experiments on pL_7 solutions at the same volume fraction as in figure 3b. DLS is a technique that allows us to probe the dynamics of our protein building blocks in solution, allowing us to extract the translational diffusion constant, D , as a function of volume fraction (Fig. S14)^{56,57}. As the volume fraction of pL_7 is increased we observe a sigmoidal transition in D from $\sim 4 \times 10^{-11} \text{ m}^2/\text{s}$ to $\sim 3 \times 10^{-11} \text{ m}^2/\text{s}$, demonstrating that above this volume fraction there is a shift in the dynamical behaviour of the protein in solution. This transition is centred at a volume fraction of $2 \pm 0.2 \%$ (Fig. S14b), which is within error of the fitted value extracted from our lag time data (Fig. 3b) and is in close agreement with the value predicted by our geometric model of 1.6% (Eqn. 1). From Doi-Edwards theory⁵⁰ we would expect a reduction in D , if the system were shifting from the dilute to the semi-dilute regime, due to a restriction in translational diffusion perpendicular to the polyprotein rod-axis⁵⁸.”*

These results demonstrate that there is a transition in the dynamics of pL_7 solutions which is responsible for a shift in the dominant network formation mechanism resulting in more rapid gelation (Fig. 3b), more homogeneous networks (Fig. 2d,e) and more rigid networks (Fig. 1b). Furthermore, this transition coincides with the value predicted by our geometric rotation model (Eqn. 1) suggesting that that shift in formation mechanism in these protein networks is from TDL to RDL. “

Q2.5 The discussion on the average coordination also raises some doubts, especially since these networks are prevalently very heterogeneous (as discussed in various parts of the manuscript) hence the average coordination cannot provide much information about rigidity. Have the authors performed rheological tests in the simulations? Or have they the possibility to establish that those are rigid in some other way?

Unfortunately, the BioNet simulations do not currently include rheological testing. We agree with the reviewer that due to the heterogeneous nature of our hydrogels the average coordination alone is not able to provide a full picture of the origin of rigidity.

We have addressed this in the manuscript by altering our language regarding the coordination, and including additional literature suggested by the reviewer (see below comment).

(Page 3) *“The low G' values at low ARs is suggest that the 1-mer and 2-mer do not form self-supporting networks. ~~ive of an under coordinated system, i.e. the average coordination, ζ , of each polyprotein building block is lower than 4, which does not form a fully self supporting gel network in an athermal 3D frictional system^{35,36} (Note: 2D thermal simulations have been performed⁴⁹~~*

~~which demonstrated that a significant increase in the modulus was still observed at $\zeta = \zeta_c$ despite the inclusion of thermal fluctuations to the simulated network). This is confirmed by the loss ratio (defined as a ratio loss modulus to the storage modulus) (Fig. S2), which is above one for low AR samples ($AR < 3$) exhibiting fluid dominated behaviour as opposed to the viscoelastic solid-like behaviour expected for a gel. For gels constructed from building blocks of AR three and greater, we observe loss ratios of approx. 0.03 demonstrating that these gels are self-supporting networks which are dominated by their elastic behaviour. A possible reason for this transition to a self-supporting network as AR is increased is that the network is under-coordinated at low ARs i.e. the average coordination, ζ , of each polyprotein building block is lower than 4, so would likely not be able to form a self-supporting gel network."~~

Q2.6 Also the relevance of rigidity criteria valid for athermal systems with frictional forces seems debatable for the protein gels. The authors may want to consider rather the literature on biopolymer and fiber networks, for example some of the works by the Koenderink group, showing how the network rigidity and mechanics may be controlled by both the geometry of the network branching and by the larger scale heterogeneities in a broad range of biopolymer networks (see for example Broedersz et al, Nature Phys. 2011 for rigidity, or Burla et al. PNAS 2020, for the network failure).

We thank the reviewer for signposting this literature and have included text discussing this literature in the manuscript.

(Page 3) ~~"Coarse-grained simulations were performed using the simulation platform BioNet39,41,42 (Methods), to investigate the coordination of network (polyprotein) building blocks of different ARs. Figure 1c shows how the average coordination of each polyprotein building block, ζ , varies as a function of AR, displaying a linear relationship. From the graph the average coordination of the simulated pL systems is below critical coordination at an aspect ratio of 1 but quickly passes critical coordination as AR is increased. These results suggest that while AR is a method of controlling the coordination of individual building blocks it is not sufficient to explain the sudden increase in G' observed when $AR = 3$. Previous literature^{43,44} on fibrous protein networks has observed that the rigidity and mechanical behaviour of heterogeneous protein networks are governed by a combination of building block coordination and connectivity (i.e. branching) between bundled protein fibres. Similarly, it has been observed by Del Gado et al.^{45,46} that in heterogeneous clustered colloidal networks that only the connections between clusters are significant in defining rigidity (i.e. the highly coordinated particles inside clusters do not contribute significantly to the rigidity). Folded protein hydrogels have been observed to exhibit highly heterogeneous clustered structures⁴⁷⁻⁴⁹, so we would expect that the formation of a self-supporting network would be governed by a combination of building block coordination and connectivity between the clusters of proteins. Furthermore, ζ only passes a critical coordination required for a self-supporting gel network, ζ_c , (i.e. when $\zeta = \zeta_c = 4$)⁴³ when the AR is greater than three. The simulation results support a view that despite the number of available crosslinking sites (4 tyrosine residues per pL domain), low AR pL building blocks form under-coordinated networks, where the number of crosslinks formed is less than the available sites. A sudden increase in G' is observed when $AR = 3$. At this point $\zeta \sim \zeta_c$, meaning that the system can form a self-supporting gel network and exhibit the viscoelastic solid behaviour expected for a gel (Fig. S2)."~~

(Page 4) "The first of these mechanisms has already been discussed above and is likely due to a combination of coordination and system branching which enables the formation of a self-

supporting network. and is due to the increase in ζ with AR, such that it crosses over the ζ_c threshold (Note: a unit increase in AR introduces another four potential cross linking sites to the building block, without increasing the total number of crosslinking sites in the system at a given protein concentration). The second mechanism..."

Reviewer #3

Q3.1 My concerns with the work lie with the robustness of conclusions gleaned from experimental data. For example, the experimental work using protein L is predicated upon the assumption that pL polyproteins with 4-aa linkers act as stiff rod-like objects. However, there is no evidence presented in the manuscript to demonstrate that this is the case. It could be possible, for example, that a 4-aa linker is sufficiently short to introduce a kink between neighbouring pL domains, so that instead of poly-pL being a straight rod, it is a bent/kinked rod. Can the authors demonstrate that their poly-pL constructs indeed are structured as straight rods? If not, can they comment on the implications of distinct structures such as a curved rod, in the context of their model?

We have provided additional SAXS data to support our assumption that protein L constructs are rigid rods. The SAXS curves of the polyprotein exhibit the characteristic ankle at high q and the q^{-1} that we expect for a rod-like object of a finite thickness. If the rods were more polymeric, we would expect a power law of 1.66 for an extended chain and 2 for a Gaussian chain.

We have added figure S2 and text to the main manuscript.

(Page 3) *"We perform small-angle x-ray scattering to confirm our rationale regarding the rigidity of our polyproteins. The scattering curve of pL₇ (our longest polyprotein, i.e. the most likely to be flexible) shown in figure S2, exhibits a Porod exponent of 1 which is indicative of a rod or a polymer in its fully extended conformation⁴⁰, which suggests that our polyproteins are rigid and can be modelled as rod-like."*

Q3.2 Furthermore, the structure of pL included in Fig. 1a seems to already have an AR greater than 1: perhaps closer to 1.5 or 2. In the clearly described BioNet simulations, pL is modelled as two connected spheres, implying an AR close to 2. Why is the AR of a single pL assumed to be 1 throughout the presentation of the experimental work? How much does this assumption affect the interpretation of results?

By comparing the N-C termini distance (The N and C termini are the linking sites between tandem repeats of protein L) of $\sim 25\text{\AA}$ to the equatorial diameter of protein L (determined in UCSF Chimera) of $\sim 22\text{\AA}$, we found the aspect ratio of protein L to be ~ 1.15 . For simplicity of modelling, we assume that the aspect ratio of pL is 1 i.e. pL is spherical. We expect this assumption to have a minimal effect on the interpretation of our results, as this would only slightly shift the extracted AR_c values from figure 1 but they would remain in reasonable agreement with equation 1. Furthermore, the predicted value of ϕ_{crit} for pL₇ would decrease to approx. 1.2% which is still within the error range of our extracted ϕ_{crit} value (from Fig. 3b) of $1.5 \pm 0.3\%$. This suggests that this small alteration in the aspect ratio would not alter our interpretation of our data.

We thank the reviewer for bringing the discrepancy between our experimental and computational modelling to our attention and believe that the error originated due to a miscommunication. We apologise for the confusion this has caused. To correct this, we have reperformed the simulations with pL modelled as a sphere and amended figure 1 to contain the new data (Please note there

was an additional error in our initial analysis which resulted in incorrect correlation, this has also been corrected). We have altered the text in methods to reflect this change.

(Page 12) *“In the present work, each protein L representative subunit was modelled as a spherical subunit of radius, $R = 15\text{\AA}$. These subunits were given 4 equally distributed binding sites and subsequently assembled into oligomers with AR values in the range of those used experimentally.”*

Q3.3 The agreement between equation (1) and the data for ARc vs concentration is striking (Figure 1d) and seems to be strong evidence in favour of the model of rotational diffusion contributing to the transition behaviour of G' . However, it is not clear how the authors have converted from concentration to phi, volume fraction. Is the conversion factor experimentally determined or found via this fit?

We have added the relevant equation to the methods and have included references for the value of protein density used ($\rho_{\text{protein}} = 1.37$).

(Page 12)

“Conversion from concentration to volume fraction

Throughout this work the concentration of both protein L and fibrinogen is converted into volume fraction and vice versa. Equation 7 shows the formula for the conversion from concentration in mg/ml to volume fraction,

$$\phi = \frac{\text{conc} \left(\frac{\text{mg}}{\text{ml}} \right)}{1000 \cdot \rho_{\text{prot}}} \quad (7)$$

where ρ_{prot} is the average density of protein (which in this work is taken to be 1.37 g/cm^3)^{63,64}.”

Q3.4 The conversion between phi and concentration seems to be a key parameter that can support the conclusions of this work, and it would be good to show whether this value is physically reasonable. From my back-of-the-envelope estimates, it appears that the phi crit value should be closer to 3.8% if ARcrit is 4.52, in contrast to the value of 1.6% provided following equation (3). It would also be good to provide the range of concentrations / phi values for which their experimental results are in the dilute regime (discussion following equation (2)). The discussion around phi and concentrations and the different regimes of dynamics for fibrin would also be assisted by providing the reader a sense of how these values are converted.

We have added the conversion equation to the methods, see above comment.

Equation 3 predicts that the critical is 1.6% as the aspect ratio of the building block considered is 7 not 4.52. If one substitutes AR = 7 into equation 1 they will obtain approximate 1.6% as the critical volume fraction for a building block of aspect ratio 7.

Q3.5 The authors also provide simulations from a kinetic lattice model as support for their rotational diffusion mechanism. The model does not include rotation and the results of the simulations show a linear, rather than sigmoidal, trend in Df. Disagreement between these results and the experimental findings does not “confirm[ing] that rotational diffusion is a key mechanism driving the architecture of protein networks”. The model is unable to confirm this, without explicitly adding rotational diffusion and showing that the sigmoidal trend in Df is recovered. I do not find that the kinetic lattice model is useful in helping to provide a mechanistic understanding

of network assembly and structure. The cited reference for more details about the model (ref 64) is also not useful at the current time.

We agree and have altered the text to reflect more accurately what the model has demonstrated. Reference 64 (now reference 76) has now been published in Soft Matter and the reference updated to reflect this. (Page 6-7) "... and 50 mg/ml. These results confirm that translational diffusion alone is not sufficient to produce the transition in network structure and thereby suggesting that rotational diffusion is a key mechanism driving the architecture of protein networks."

Q3.6 The storage modulus as a function of AR is clearly not well described as a sigmoid. However, the use of a double sigmoid function is concerning, given the number of fit parameters (6) and data points (7). With so many degrees of freedom, it should be easy to fit their data to this function. At a minimum, the authors should report statistical goodness of fit parameters such as reduced chi squared.

Protein engineering dictates that we are unable to produce polyproteins with "fractional-domains" hence we cannot obtain non-integer values for the aspect ratio experimentally.

We have added text caption in figure 1 and to the supplementary info. Additionally, have provided chi-squared values in supplementary table S2.

(Figure 1 caption, Page 4) "...fit (Eqn. 4), full fitting parameters provided in Table S2. Dashed..."

(Figure S5 caption) "a single sigmoid function (red, reduced χ^2 : 21) and a dual sigmoid function (green, reduced χ^2 : 0.33)."

Minor points:

What do the error bars in Figure 1b represent, and how many replicates were performed for each measurement?

The error bars in figure 1b represent the fitting error of the fractal dimension. Due to the nature of neutron scattering experiments including the limited time and volume of sample required (> 500ul per sample) we were unable to perform SANS on replicates.

We have added text to the caption of figure 2 to make this clear to the reader.

(Figure 1 caption, Page 4) "...and 50mg/ml (dark purple). The error bars here denote the fitting error of the fractal dimension to SANS curves. Solid lines..."

What aspect ratio are the networks in the insets to Figure 1c meant to depict?

The insets in figure 1 depicts an object of aspect ratio one forming a network, and was intended only to simply demonstrate the difference between an undercoordinated and self-supporting network. In order to avoid confusion we have removed these cartoon schematics from figure 1.

It is unclear what the star-like objects in the schematic insets of Figure 1d are meant to represent.

These star-like object were intended to depict collision; however, we have removed them to avoid confusion.

I am not sure what the importance/relevance of the following sentence is. I suggest clarifying or eliminating, to avoid confusion. "Note: 2D thermal simulations have been performed which

demonstrated that a significant increase in the modulus was still observed at $\zeta = \zeta_c$ despite the inclusion of thermal fluctuations to the simulated network.”

We have included this sentence and reference, to note for the reader that while the critical coordination of 4 only holds for an athermal system, thermal simulations have been performed which observe a critical coordination very close to 4 so is a relevant limit to consider for our system.

We have edited the main text to improve clarity as requested

(Page 2) “(Note: 2D thermal simulations have been performed³⁷ which demonstrated that a significant increase in rigidity of the simulated network was observed at critical coordination despite the inclusion of thermal fluctuations).”

Why are the data points for D_f vs AR in Fig 2d fit to a sigmoid model only for 50 mg/ml? It looks like a sigmoid could also describe the 25 mg/ml data.

We have fit a sigmoid to the 25mg/ml D_f vs AR data sets and added to the main text.

(Page 6) “...of 4.52. Additionally, we also fitted a sigmoid to the D_f results at 25 mg/ml and find an $AR_{midpoint}$ of 6.5 ± 0.8 , which also agrees with AR_{crit}^{rod} at 25 mg/ml (predicted with Eqn. 1) of 6.37. These results suggest...”

“It is intuitive that higher AR building blocks will link together to form much more porous structures”. My intuition could tell me the opposite: that higher AR building blocks can pack laterally, as is the case in collagen fibrils, and make structures whose backbones are highly dense and aligned. Can this statement in the manuscript be clarified?

The maximum packing efficiency of any cylinder regardless of AR is approximately 0.9, so increasing AR makes no difference to the maximum packing of cylinders. However, we agree we have not been clear enough and have added text to clarify that we mean randomly assembling sticky rods are intuitively more likely to form more porous structures.

(Page 6) “This result is to be expected as the D_f is a measure of the space-filling capacity of an object, and it is intuitive that sticky higher AR building blocks that are diffusing and linking together randomly will form much more porous structures (due to the elongated dimension).”

It was only in the methods section that I realized that gel formation was triggered by light exposure. It could be helpful to add this earlier in the manuscript.

We have highlighted this more clearly earlier in the manuscript as requested.

(Page 2) “Tyrosine residues are essential for the residue specific photo-chemical crosslinking method that we employ in this work to form our pL hydrogel networks.”

All parameters from fits to data should be provided for the reader (e.g. in a table in the SI). These include all six parameters from each double-sigmoid fit.

We have provided all fitting parameters in table S2 in the SI as requested along with goodness of fit values (reduced chi-squared).

It is not clear to me how rates influence G' as measured in Figure 1b. It is also not clear why this rate * AR gives a unitless number. Please clarify the basis for the form of equation (4).

The basis of equation 4 is two sigmoids added together where k_z and k_c define the effective 'width' of the transition region. We have altered to the text to remove the word rate in order to not cause confusion.

(Page 11) *"...finally k_z and k_c are related width of the transitions centred around AR_z and AR_c , respectively, at which G' increases to plateau when the system passes critical coordination and rotational limit,..."*

The loss ratio is not defined in the text.

Have define loss ratio in the text as requested.

(Page 3) *"... This is observed in the loss ratio (defined as a ratio loss modulus to the storage modulus) (Fig. S2), which is above ..."*

The abstract states that network formation is more rapid with high AR. I may have missed it, but I do not see this explicitly addressed in the text. When I look at Figure 3, I see that pL provides a much shorter lag time than fibrin, although it has a shorter AR. Please check this.

High AR allows for more rapid formation as increased AR lowers the critical volume fraction for the transition from TDL to RDL.

REVIEWER COMMENTS

Reviewer #1 (Remarks to the Author):

The authors have taken care of majority of the comments. However, the hydrogel water uptake part is not addressed, as authors respond that it is not focus of this study. However the authors need to include this key information as the water uptake controls all the behaviour. The hydrogels normally will exhibit characteristics based on water content.

Reviewer #2 (Remarks to the Author):

I find that the authors have satisfactorily addressed the concerns I had raised in the revised manuscript and in their answer. Overall the manuscript has certainly improved and possible weak points have been discussed adequately. I am happy to recommend publication.

Reviewer #3 (Remarks to the Author):

The authors have satisfactorily addressed all of my comments in their revisions to the manuscript.

12th June 2023

Dear Reviewers

Thank you for the reviews of our manuscript.

Yours faithfully,

Professor Lorna Dougan

Please find below the reviewers comments in *italics* and our response in **bold**.

Reviewer #1:

The authors have taken care of majority of the comments. However, the hydrogel water uptake part is not addressed, as authors respond that it is not focus of this study. However the authors need to include this key information as the water uptake controls all the behaviour. The hydrogels normally will exhibit characteristics based on water content.

The goal of the current understudy was to understand the impact of protein building block aspect ratio on network formation in a hydrogel. In order to achieve this goal it is essential that the protein volume fraction (and therefore water volume fraction) is a known, fixed and unchanging quantity for all systems studied. Our approach ensured all network formation studies were completed at a known, fixed and unchanging water volume fraction and this allowed for direct comparison across the different aspect ratio systems. We have added additional text in the manuscript to clarify this.

For this reason, water update experiments were not completed and are not relevant for the subject of the current manuscript.

(Page 3) "To ensure the effects we observe are due to the building block aspect ratio and not the uptake of additional water we ensure that the protein volume fraction/concentration in mg·ml⁻¹ (and therefore water volume fraction) is a known fixed value at each protein concentration tested (see Methods)."

(Page 11) "To prevent evaporation and water uptake, during the rheology measurements low viscosity silicone oil (approximately 5 ct) was placed around the sample and geometry."

(Page 12) "To prevent exchange of deuterium with hydrogen in the air, evaporation and water uptake the quartz cuvettes were sealed with parafilm."

Reviewer #2:

I find that the authors have satisfactorily addressed the concerns I had raised in the revised manuscript and in their answer. Overall the manuscript has certainly improved and possible weak points have been discussed adequately. I am happy to recommend publication.

Thank you for this feedback.

Reviewer #3:

The authors have satisfactorily addressed all of my comments in their revisions to the manuscript.

Thank you for this feedback.

REVIEWER COMMENTS

Reviewer #1 (Remarks to the Author):

As per the response and the paper, I do not see anything included for a simple basic question of water content in the hydrogel. The whole study is pivoting around hydrogel. The author should include the volume fraction of water present in the network. Whether it is not the aim or goal of the project, it is crucial that author take utmost care to report the very basic thing about this hydrogel. One must know the water content or any equilibrium water content in the gel. All rheological data will be influenced by the water content. The authors themselves mentioned "To prevent evaporation and water uptake, during the rheology measurements low viscosity silicone oil (approximately 5 ct) was placed around the sample and geometry".

Why the authors cannot include the water amount present in the samples for which they did rheology, SANS with D2O. They can do a simple TGA experiment on small amount of hydrogel sample to report the amount of water present.

5th July 2023

Dear Reviewers

Thank you for the reviews of our manuscript.

Yours faithfully,

Professor Lorna Dougan

Please find below the reviewers comments in *italics* and our response which follows.

Reviewer #1:

As per the response and the paper, I do not see anything included for a simple basic question of water content in the hydrogel. The whole study is pivoting around hydrogel. The author should include the volume fraction of water present in the network. Whether it is not the aim or goal of the project, it is crucial that author take utmost care to report the very basic thing about this hydrogel. One must know the water content or any equilibrium water content in the gel. All rheological data will be influenced by the water content. The authors themselves mentioned "To prevent evaporation and water uptake, during the rheology measurements low viscosity silicone oil (approximately 5 ct) was placed around the sample and geometry". Why the authors cannot include the water amount present in the samples for which they did rheology, SANS with D2O. They can do a simple TGA experiment on small amount of hydrogel sample to report the amount of water present.

We apologise for the confusion we seem to have caused.

We have included two supplementary tables in which we list both the **protein and water volume fraction of each hydrogel sample**. We have signposted to these tables in the captions of the relevant figures.

25 mg·ml ⁻¹			37.5 mg·ml ⁻¹			50 mg·ml ⁻¹		
pL Aspect Ratio	Protein Volume Fraction (%)	Water Volume Fraction (%)	pL Aspect Ratio	Protein Volume Fraction (%)	Water Volume Fraction (%)	pL Aspect Ratio	Protein Volume Fraction (%)	Water Volume Fraction (%)
1	1.82	98.18	1	2.74	97.26	1	3.65	96.35
2	1.82	98.18	2	2.74	97.26	2	3.65	96.35
3	1.82	98.18	3	2.74	97.26	3	3.65	96.35
4	1.82	98.18	4	2.74	97.26	4	3.65	96.35
5	1.82	98.18	5	2.74	97.26	5	3.65	96.35
6	1.82	98.18	6	2.74	97.26	6	3.65	96.35
7	1.82	98.18	7	2.74	97.26	7	3.65	96.35

Table S3: The protein and water volume fraction of protein L construct hydrogels as a function of pL construct aspect ratio and at varying protein concentrations in mg·ml⁻¹, equivalent to those show in figure 1b,d and 2d.

Protein L 7-Mer			Fibrinogen		
Protein Conc (mg·ml ⁻¹)	Protein Volume Fraction (%)	Water Volume Fraction (%)	Protein Conc (mg·ml ⁻¹)	Protein Volume Fraction (%)	Water Volume Fraction (%)
6.25	0.46	99.54	0.1	0.01	99.99
12.5	0.91	99.09	0.23	0.02	99.98
25	1.82	98.18	0.5	0.04	99.96
37.5	2.74	97.26	0.75	0.05	99.95
50	3.65	96.35	1	0.07	99.93
			1.5	0.11	99.89
			2.2	0.16	99.84

Table S4: The concentrations in mg·ml⁻¹ of Protein L 7-mer and Fibrinogen used in the lag time measurements (Fig. 3), with the corresponding protein and water volume fraction.

Additionally, we have included text in the methods section which outlines how the protein and water volume fraction are calculated from the empirically measured protein concentration in mg·ml⁻¹.

(Page 4) "...concentration. Table S3 shows the equivalent protein and water volume fractions of pL hydrogel. c) Average..."

(Page 5) "...respectively. Table S3 shows the equivalent protein and water volume fractions of pL hydrogel. e) Schematic..."

(Page 8) "...networks. Table S4 show the protein and water volume fractions of each pL7 lag time samples. c) Schematic..."

(Page 8) "...networks. Table S4 show the protein and water volume fractions of each of the fibrinogen lag time samples. Solid..."

(Page 12) "*Conversion from concentration to volume fraction*

Throughout this work the concentration of both protein L and fibrinogen is converted into volume fraction and vice versa. Equation 7 shows the formula for the conversion from protein concentration in mg·ml⁻¹ to protein volume fraction,

$$\phi = \frac{\text{conc} \left(\frac{\text{mg}}{\text{ml}} \right)}{1000 \cdot \rho_{\text{prot}} \left(\frac{\text{g}}{\text{cm}^3} \right)} \quad (7)$$

where ρ_{prot} is the average density of protein (which in this work is taken to be 1.37 g·cm⁻³)^{75,76}. The water content of hydrogels, ϕ^{water} , is calculated by taking the protein volume and subtracting it from unity, $\phi^{\text{water}} = 1 - \phi$. Note that the protein concentration in mg·ml⁻¹ is determined via the absorption at 280nm in conjunction with the Beer-Lambert Law."

REVIEWERS' COMMENTS

Reviewer #1 (Remarks to the Author):

In the revised version, the authors provided a list of both the protein and water volume fraction, which are used for preparing the hydrogel sample. However, the query is related to water content in the final hydrogel network (e.g PL hydrogel after crosslinking with the Ru-bi pyridyl, i.e. the hydrogel network). The initial concentration (mg/ml) reported in the revised version of the manuscript table does not represent the water content in the hydrogel network.

What is the water content in the crosslinked hydrogel? Have the authors done any experiment with drying and reswelling of the gels to report the water uptake or water content in the gel.

31st July 2023

Dear Reviewers

Thank you for the reviews of our manuscript.

Yours faithfully,

Professor Lorna Dougan

Please find below the reviewers comments in *italics* and our response which follows.

Reviewer #1:

In the revised version, the authors provided a list of both the protein and water volume fraction, which are used for preparing the hydrogel sample. However, the query is related to water content in the final hydrogel network (e.g PL hydrogel after crosslinking with the Ru-bi pyridyl, i.e. the hydrogel network). The initial concentration (mg/ml) reported in the revised version of the manuscript table does not represent the water content in the hydrogel network.

What is the water content in the crosslinked hydrogel? Have the authors done any experiment with drying and reswelling of the gels to report the water uptake or water content in the gel.

The water content in the cross-linked hydrogel is provided in supplementary tables 2 and 3. The initial water content and final water content are the same with no water uptake or evaporation. The water content is unchanged throughout the experiments. This allows for direct comparison of proteins with different building block aspect ratios and their impact on protein network formation.

A known and consistent protein concentration in solution is crucial for determining the interplay between the protein aspect ratio and volume fraction in the pre-gel solution and its role in defining the translational or rotational limited network formation. Drying or reswelling experiments on the gels would alter the volume fraction of protein in the network and we would not be able to complete our comparative study of AR building blocks at a fixed volume fraction.

To make the discussion above clear to readers we have added additional text to the results and discussion sections (including additional references).

(Page 7) "...mechanics of hydrogel networks. By increasing the AR of protein building blocks through the inclusion of additional pL domains and comparing different AR proteins at a fixed protein volume fraction, we have demonstrated that the mechanical rigidity (G') of the network increases with increasing AR up to a plateau value. A fixed and known protein, and therefore water, volume fraction is crucial to allow for comparison of protein network formation across the different AR building blocks. This is distinct from previous studies which have studied the mechanical properties of the swollen hydrogels after network formation and relaxation in a specific buffer of interest, to exploit chain entanglement for creating cartilage-like material properties⁶⁶ and bi-layer swelling to engineer shape memory and morphing biomaterials⁶⁷."

And earlier in the manuscript

(Page 3) "...material present in the system. To allow for comparison of proteins with different building block aspect ratios and their impact on protein network formation the protein volume fraction and water volume fraction are a known fixed value and are unchanged throughout the experiments. The protein volume fraction/concentration in $\text{mg}\cdot\text{ml}^{-1}$ (and therefore water volume fraction) is the same at the start and end of the experiment, with no water uptake or evaporation (see Methods). The protein and water volume fraction are listed in supplementary tables 2 and 3. The low G' values..."